# The influence of the synoptic regime on stable water isotopes in precipitation at Dome C, East Antarctica

**Elisabeth Schlosser[1,2], Anna Dittmann[1], Barbara Stenni[3],  Jordan G. Powers[4], Kevin W. Manning[4], Valérie Masson-Delmotte[5], Mauro Valt[6], Anselmo Cagnati[6], Paolo Grigioni[7] and Claudio Scarchilli[7]**

[1]Inst. of Atmospheric and Cryospheric Sciences, University of Innsbruck, Innsbruck, Austria

[2]Austrian Polar Research Institute, Vienna, Austria

[3]Department of Environmental Sciences, Informatics and Statistics, Ca 'Foscari University of
    Venice, Venice, Italy

[4]National Center for Atmospheric Research, Boulder, CO, USA

[5]Laboratoire des Sciences du Climate et de l'Environnement, Gif-sur-Yvette, France

[6]ARPA Center of Avalanches, Arabba, Italy

[7]Laboratory for Observations and Analyses of the Earth and Climate, ENEA, Rome, Italy

Submitted to:  The Cryosphere

13 February 2017

Revised version: 10 May 2017

re-revised version: 30 August 2017

final version: 12 September 2017

*Correspondence to*: Elisabeth Schlosser (Elisabeth.Schlosser@uibk.ac.at)

**Abstract.** The correct derivation of paleotemperatures from ice cores requires exact knowledge of all processes involved before and after the deposition of snow and the subsequuent formation of ice. At the Antarctic deep ice core drilling site Dome C, a unique data set of daily precipitation amount, type and stable water isotope ratios is available that enables us to study in detail atmospheric processes that influence the stable water isotope ratio of precipitation. Meteorological data from both automatic weather station and a mesoscale atmospheric model were used to investigate how different atmospheric flow patterns determine the precipitation parameters. A classification of synoptic situations that cause precipitation at Dome C was established and, together with back-trajectory calculations, was utilized to estimate moisture source areas. With the resulting source area conditions (wind speed, sea surface temperature (SST) and relative humidity) as input, the precipitation stable isotopic composition was modelled using the so-called Mixed Cloud Isotope Model (MCIM). The model generally underestimates the depletion of $^{18}$O in precipitation, which was not improved by using condensation temperature rather than inversion temperature. Contrary to the assumption widely used in ice core studies, a more northern moisture source does not necessarily mean stronger isotopic fractionation. This is due to the fact that snowfall events at Dome C are often associated with warm air advection due to amplification of planetary waves, which considerably increases the site temperature and thus reduces the temperature difference between source area and deposition site. In addition, no correlation was found between relative humidity at the moisture source and the deuterium excess in precipitation. The significant difference in the isotopic signal of hoarfrost and diamond dust was shown to disappear after removal of seasonality. This study confirms the results of an earlier study carried out at Dome Fuji with a shorter data set using the same methods.

## 1 Introduction and previous work

### 1.1 Ice cores in paleoclimatology

Ice cores from the vast ice sheets of Greenland and Antarctica have proven to be of high value in paleoclimate research. Of particular importance is the use of stable water isotope ratios as proxy for deriving past temperatures. However, it has been shown that the calibration of the "paleothermometer" is not as straightforward as originally assumed. Various factors apart from air temperature influence the stable isotope ratio, both before and after the deposition of the snow that develops into ice by metamorphosis. Post-depositional processes were thought

to occur mainly within the snow pack, firn or ice. Recent studies have shown, however, that the interaction between the uppermost layers of the snowpack and the overlying atmosphere between precipitation events also plays an important role. This was found in both Greenland

(Steen-Larsen et al., 2013; Bonne et. al., 2013) and Antarctica (Casado et al., 2016a, 2016b; Ritter et al., 2016; Touzeau et al., 2016) as well as in laboratory experiments (Ebner et al., 2017).

In this study we focus on the processes before deposition, namely atmospheric processes related to moisture transport and precipitation formation. The precipitation data used here

enable us to exclude post-depositional processes to study the purely atmospheric influence on precipitation. Since the stable water isotope ratio changes during evaporation and condensation processes (Dansgaard, 1964), it is important to know as much as possible about the history of the precipitation observed at an ice core drilling site, specifically moisture source, moisture transport paths, and meteorological conditions at both the moisture source

and the deposition site. Precipitation measurements in Antarctica are rare due to the large technical difficulties of measuring precipitation at extremely low temperatures or high wind speeds. However, at the deep-drilling location Dome C on the East Antarctic plateau, a series of precipitation data has been collected that includes not only precipitation amounts but also precipitation type and stable isotope ratios. This unique data set can be combined with a full

meteorological data set including radiosonde data, AWS data, and atmospheric model data. This, for the first time, allows us to study in detail the synoptic conditions that lead to precipitation at Dome C and how they are related to the precipitation stable isotope ratios. We compare our results to those of a similar study carried out by Dittmann et al. (2016) for Dome Fuji, Dronning Maud Land, another deep ice core drilling site, where a one-year series

of combined stable isotope and precipitation data is available. In both studies exactly the same methods were used for calculation of transport pathways and isotopic fractionation as well as for synoptic analysis, which is highly valuable as often past studies have site specific approaches, making comparisons very challenging.

**1.2 Stable isotopes**

Since the ground-breaking work of Dansgaard (1964), stable water isotopes have become one of the most important parameters measured in ice cores. An empirical linear relationship was found between the annual mean air temperature (derived from the 10m-snow temperature)

and the annual mean $\delta^{18}O$ of snow samples along traverses in Antarctica and Greenland (Jouzel et al., 1997). However, it became clear fairly early that this spatially derived relationship was different from the corresponding temporal relationship and thus could not be used as calibration for calculating paleotemperatures from ice core stable isotope ratios (e.g. Masson-Delmotte et al., 2008). More recently, it has been found that the temporal relationship is not constant for different climates or even time periods within a glacial or interglacial (Sime et al., 2009). Spatial differences in the temporal relationship are common, and the relationship can vary with season (e.g. Kuettel et al., 2012). While the empirical equation relates the stable isotope ratio only to the condensation temperature at the deposition site, various other factors influence this ratio, such as moisture source conditions and vertical and horizontal transport paths, entrainment of additional moisture along the way, sea ice conditions, seasonality and intermittency of precipitation as well as postdepositional processes. The second-order parameter deuterium excess (d=$\delta D$-8*$\delta^{18}O$), which combines the information from $\delta^{18}O$ and deuterium, has been used to derive information about both condensation temperature and moisture source conditions, namely wind speed, SST, and relative humidity (e.g. Stenni et al., 2001, Uemura et al., 2012)). Most recently, due to the development of new measuring techniques, the rare isotope $^{17}O$ and the corresponding $^{17}O$-excess have been introduced into ice core studies (e.g. Landais et al., 2008, 2012: Schoenemann et al., 2014). The $^{17}O$-excess is supposed to be insensitive to evaporation temperature and less sensitive than d-excess to equilibrium fractionation processes during formation of precipitation. Thus it may offer the potential of disentangling the different effects of fractionation during evaporation, moisture transport, and precipitation formation (Schoenemann et al., 2014).

A variety of models is used to simulate isotopic fractionation, from simple Rayleigh-type distillation models to fully three-dimensional atmospheric circulation models. So far, most models are still based on the early theories developed by Jouzel and Merlivat (1984). Ciais et al. (1994) extended this theory to mixed clouds in their Mixed Cloud Isotope Model (MCIM), which is described further in the methods section.

Kavanaugh and Cuffey (2003) developed a model of intermediate complexity (ICM), more complex than simple Rayleigh-type models, but not as sophisticated as General Circulation Models (GCM), to study how variations in single climate parameters or in fundamental characteristics of isotopic distillation affect the stable isotope ratio of polar precipitation. Schoenemann and Steig (2016) applied their model to $^{17}O$-excess, using data from Vostok and the WAIS core for comparison. GCMs are so far not able to correctly represent d-excess or $^{17}O$-excess measured at Dome C (Stenni et al., 2010). In a most recent study, Steen-Larsen et

al. (2017) evaluated various isotope enabled GCMs against in situ atmospheric water vapour isotope measurements. They found that, apart from a poor performance of all models for d-excess, biases in $\delta^{18}O$ could not be explained simply by model biases in air temperature and humidity.

In the discussion of sea level rise, often the possibility of a mitigation of sea level rise by increased Antarctic precipitation, the most important component of the surface mass balance (SMB) is mentioned (e.g. Church et al, 2013). However, the relationship between stable isotope ratios and precipitation/accumulation is yet fully understood. Most commonly, the assumption of a positive correlation between stable isotope ratio (as proxy for air temperature) and accumulation rate has been used based on the relationship between temperature and saturation water vapour pressure (Clausius-Clapeyron). However, contrasting results are found in the recent literature. While Frieler et al. (2015), using both model and ice core data, state that Antarctic accumulation increases with rising air temperature, Fudge et al. (2016) found that the relationship between accumulation and temperature has not been constant over the past 30000 years in West Antarctica. They stated that atmospheric dynamics play a more important role than thermodynamics, which had also been found by Altnau et al. (2015) and Schlosser et al. (2014) in coastal Dronning Maud Land.

## 1.3 Synoptic analysis

In the past, precipitation in the interior of the Antarctic continent was only poorly understood, because only a few meteorological observatories have existed in continental Antarctica. A analysis of satellite imagery has brought only limited progress due to the difficulty of distinguishing between clouds and the snow surface (Massom et al., 2004). Since the improvement of global and mesoscale atmospheric models, however, our knowledge has advanced considerably. Noone et al. (1999) studied precipitation conditions in Dronning Maud Land (DML) using ECMWF reanalysis data. They found that 89 % of the days have low precipitation,(<0.2 mm/d) corresponding to 31 % of the annual total, whereas only 1 % of the days have high precipitation (>1mm/d), but that the latter account for 20 % of the annual precipitation. High precipitation days were shown to be connected to amplified upper level planetary waves that direct moist air towards DML. Various studies have confirmed and extended these results for different parts of Antarctica. For example, it has been shown that a few snowfall events per year can be responsible for up to half of the annual total precipitation

(Braaten, 2000, Reijmer and Van den Broeke, 2003; Fujita and Abe, 2006; Schlosser et al., 2010a, Gorodetskaya et al., 2013). In addition, Gorodetskaya et al. (2014) showed that atmospheric rivers play an important role in heavy precipitation events in Antarctica.

Synoptic events with blocking anticyclones were also described by Scarchilli et al. (2010), Massom et al. (2004), and Hirasawa et al. (2000). At the deep-drilling site Dome Fuji, while warm air advection combined with orographic lifting sometimes was not sufficient for precipitation formation, it did cause the removal of the prevalent temperature inversion layer by cloud formation that increased the downward long-wave radiation and by turbulent mixing
(Enomoto et al. 1998; Hirasawa et al., 2000). Also, increased amounts of diamond dust can be observed after a synoptic snowfall event when moisture levels are still higher than on average (Hirasawa et al., 2013; Dittmann et al., 2016; Schlosser et al., 2016).

Dittmann et al. (2016) analysed the only other daily precipitation/stable isotope ratio data set available in the interior of Antarctica, which was created in 2003 at the Japanese deep drilling
site Dome Fuji (Fujita and Abe, 2006). They investigated synoptic conditions during precipitation, estimated moisture source areas for precipitation events and used MCIM to model the stable isotope ratios. Five typical weather situations for precipitation were defined. Approximately two thirds of the days were directly or indirectly related to advection of moist air associated with amplification of the planetary waves. The model represented the observed
annual cycle of $\delta^{18}$O and deuterium excess fairly well, but it underestimated the amount of fractionation between first evaporation at the oceanic moisture source and deposition at Dome Fuji. Nicolas and Bromwich similarly documented intrusions of warm maritime air into West Antarctica (Nicolas and Bromwich, 2011). Schlosser et al. (2004) investigated the influence of origin of precipitation on the delta-T relationship using a 20yr series of fresh snow samples
at Neumayer Station, coastal Dronning Maud Land (DML). They calculated backward trajectories for three different arrival levels and compiled a classification of synoptic situations related to the precipitation events. The quality of the delta-T relationship varied for the different trajectory classes (i.e. the differing moisture origins), and significant differences were found in both the delta-T slopes and the deuterium excess for the different classes.

For ice core interpretation, these findings are important since they contradict the older assumption that precipitation in the interior Antarctica is predominantly diamond dust and thus exhibits only a weak seasonality. This implies that all seasons are represented evenly in the ice core. If, however, the synoptic snowfall occurs preferably in certain seasons and/or this preference is not constant in different climates, potentially a cold or warm bias would be
found in the temperatures derived from stable water isotopes of an ice core. An understanding

of the atmospheric circulation and its influence on precipitation conditions at deep drilling sites is therefore essential for a correct interpretation of the ice core proxy data.

For Antarctica, only very few studies exist, that combine daily precipitation/fresh snow stable isotope data with meteorological conditions at the time of precipitation. The study at hand is the first to use a multi-year time series of such data for a deep ice core drilling site in the interior of the Antarctic continent.

## 2 Study site

Dome C (75.106 °S, 123.346 °E) is one of the major domes in East Antarctica, at an elevation of 3233 m. Since 2005, a wintering base has been operated there jointly by France and Italy ("Concordia Station"). Dome C has a mean annual temperature of -54.5 °C (derived from 10m-snow temperature; mean temperature from AWS 1996-2015: -51.3°C) and a mean annual accumulation of 25 mm water equivalent (w.e.), the latter derived from ice cores. Dome C is the site where the so far oldest ice has been retrieved during the European Project for Ice Coring in Antarctica (EPICA). After the first core, with a depth of 906 m covering ca. 32 000 years, had been drilled in 1977/78 (Lorius et al., 1979), several cores followed, and in January 2006, the EPICA drilling was completed at a depth of 3270.2 m, yielding ice approximately 800 000 years old. This core thus covers eight glacial cycles (EPICA community members, 2004), which doubles the time span that had been represented in ice cores previously.

## 3 Data and methods

### 3.1 Precipitation and stable isotope data

Precipitation has been measured and sampled at Dome C since 2006 (with some interruptions in the early time period) and this sampling is ongoing. A wooden platform of approximately 1m height, covered by a polystyrene/teflon plate is used to measure daily precipitation amounts. The elevated platform is surrounded by a rail of 5 cm height. This helps to avoid contributions from low drifting snow, but cannot prevent that precipitated snow is removed completely - and thus not measured - at higher wind speeds. The platform is located at a distance of 800 m from the main station. Until the end of 2007, the measurements were not

carried out daily, and the samples were collected only when precipitation reached a certain threshold, which led to sampling intervals of four to five days. Since December 2007, however, precipitation has been sampled once per day at 0100 UTC, and amounts and stable isotope ratios of the samples are determined. In this study we therefore consider the time period 2008 – 2010, with the more recent samples having not yet been analysed at the time of our study. For this period, measurable precipitation was observed on 59 % of all days, stable isotope ratios were determined on 45 % of the days.

Furthermore, the crystal type of the precipitation is analysed, so that diamond dust, drift snow and regular snowfall can be distinguished. Diamond dust forms due to radiative cooling of almost saturated air and consists of very fine needles. Mixing of a warmer, moister air mass with cold air can also lead to supersaturation of the cold air and consequent ice crystal formation. Synoptic snowfall is marked by various types of snow crystals that depend mainly on air temperature during crystal formation, whereas drift snow can be recognised by broken crystals. Also, a mixing of crystal types can be observed. Note that the precipitation amounts are so small that errors in quantification can amount to 100 % or more. However, usually cases of diamond dust exhibit amounts one order of magnitude smaller than synoptic snowfall events. To date, the Dome C precipitation data series, complemented by stable isotope measurements, represents the first and only multi-year precipitation series at an Antarctic deep-drilling site.

We note that $\delta^{17}O$ and $^{17}O$-excess have been determined for a part of the samples, but the amount of data is not sufficient yet to get statistically significant results, and thus they are not used in the present study.

A detailed description of the measurements and a first analysis of the stable isotope data is provided by Stenni et al. (2016).

## 3.2 AWS and radiosonde data

Radiosonde data from the meteorological station at Dome C are used to determine the temperature at both the top of the surface inversion layer and the condensation level. The upper-air data are provided by the Meteo-climatological Observatory of the Italian Antarctic Research Program (PNRA). Since the beginning of the measurements in 2005, a radiosonde has been launched every day at 12 UTC, unless excessive wind speeds preventit. For each standard pressure level, geopotential height, air temperature, humidity and wind are measured,

and the data are delivered as TEMP files to the WMO (World Meteorological Organisation) Global Telecommunication System (GTS).

The current Automatic Weather Station (AWS), named Dome C II, has been installed by the Antarctic Meteorological Research Center (AMRC) in 1995. The AMRC and AWS Program are sister projects of the University of Wisconsin-Madison, which are funded by the United States Antarctic Program (USAP). USAP provides real-time and archived weather observations and satellite imagery and supports a network of AWS across Antarctica. At the

AWS, standard meteorological variables, namely air temperature, surface pressure, wind speed and direction, and humidity are measured. AWS data can be found at http://amrc.ssec.wisc.edu.

**3.3 AMPS archive data and trajectory calculation**


The Antarctic Mesoscale Prediction System (AMPS) (Powers et al. 2012) is a real-time numerical weather prediction system run to provide guidance for the weather forecasters of the United States Antarctic Program (USAP). It has been operated by the National Center for Atmospheric Research (NCAR) in support of the USAP since 2001, at first employing the

polar version of the Fifth-Generation Pennsylvania State University/NCAR Mesoscale Model (Polar MM5). Since 2006 AMPS has used the Weather Research and Forecasting (WRF) Model. The performance of WRF in AMPS and in Antarctica has been verified in a number of previous studies (see e.g. Bromwich et al. 2005; Bromwich et al. 2013; Deb et al. 2016), while model output from AMPS has supported various Antarctic investigations (e.g., Powers,

2007; Nigro et al., 2011; 2012). The AMPS archive is the repository of gridded output from AMPS from over the years (Powers et al., 2012), and WRF gridded output from the archive has supported numerous studies (Seefeldt and Cassano, 2008; Schlosser et al., 2010a; Seefeldt and Cassano, 2012; Schlosser et al., 2016). Here, AMPS archive data from the period 2008–2010 are used here in analyses of the meteorological conditions affecting Dome C and its

precipitation.

For the period analysed in this study, the AMPS WRF configuration consisted of a nested domain setup with grids of 45-km and 15-km horizontal spacing extending from the Southern Ocean poleward and covering the Antarctic continent, respectively. As the 15-km domain includes Dome C, it is the output from this grid that is used for the analyses here. Vertical

resolution in WRF for the study period reflected 44 levels from the surface to 10 hPa. The use of the AMPS archive data follows the methodology of a number of published studies

analysing conditions and regimes at ice core drilling sites across Antarctica (Schlosser et al., 2008; Schlosser et al., 2010b; Schlosser et al., 2016; Dittmann et al., 2016).

In this study, AMPS archive data are utilized to investigate the synoptic situation that lead to
precipitation and to estimate moisture sources for the precipitation events. Fully three-dimensional 5-day back trajectories were calculated with the RIP4 software (Stoelinga, 2009) and together with 500 hPa geopotential fields were used to estimate the moisture source. Conditions at the moisture source are then derived as input for the stable isotope modelling Trajectories were calculated for three different arrival levels: 300hPa, 500hPa, and 600hPa.
Since Dome C is situated at an elevation of 3233m with a monthly mean surface pressure varying between 630 and 650hPa with daily values being considerably lower (lowest observed surface pressure: 603.6hPa), the 600hPa level is the lowest standard pressure level that is always above the surface. For this location, it thus represents the flow close to the surface. 300hPa/trajectories are not shown here, since it was found that the moisture content
at this level is already too low for producing any significant precipitation and the back-trajectories hardly ever reached heights close to sea level. In this study, 500hPa is assumed to best represent the general atmospheric flow and synoptic-scale moisture transport.

**3.4 MCIM**


The so-called Mixed Cloud Isotope Model (MCIM) is a simple Rayleigh-type model that, however, allows the co-existence of water droplets and ice crystals and, as such, is the consequent further development of the basic distillation model established by Jouzel and Merlivat (Jouzel and Merlivat, 1984; Merlivat and Jouzel, 1979). It is still widely used in ice
core studies and also is the basis for implementation of stable isotopes in General Circulation Models (GCM) or climate models. The model calculates fractionation in an isolated air parcel between the initial evaporation and the final precipitation. In contrast to a pure Rayleigh model, an adjustable part of the condensate stays in the cloud. In a likewise adjustable range of temperatures, both liquid droplets and ice crystals occur in the cloud, which causes
additional kinetic fractionation processes due to the Bergeron-Findeisen effect: because of the different saturation vapor pressure with respect to ice and water, the actual vapor pressure lies between the saturation vapor pressure above water and that above the ice. This means a sub-saturated environment for liquid water but a supersaturated environment for ice. This results in a net transport of water vapour from the droplets to the ice, with fractionation during
evaporation from the droplets and deposition (i.e. negative sublimation) on the ice crystals.

No fractionation is associated with freezing of liquid droplets since the freezing is rapid (Ciais and Jouzel, 1994) . The initial isotopic composition of the vapor after the first evaporation is calculated assuming a balance between evaporation and condensation. Details about MCIM can be found in Ciais and Jouzel (1994) and Dittmann et al. (2016).


## 4 Results

### 4.1 Meteorological conditions  at Dome C

Figure 1 shows a histogram of daily precipitation amounts at Dome C for the period 2008-2010 derived from a) measurements and b) AMPS archive data. It shows a positively skewed distribution: in both model and observations, a large number of extremely small amounts are observed compared to only a few events with more than 0.2 mm. The 90 % and the 95 % percentiles are shown as possible thresholds for synoptically caused snowfall events. Note
that the observational data refer basically to the precipitation sampling and cannot be corrected for cases where the wind speed was so high that no sampling was possible because the snow had been scoured from, or not accumulated at all, upon the platform. The small amounts most likely are associated with diamond dust formation, whereas the larger events are related to synoptically caused snowfall events (hereafter called "synoptic snowfall"
events), which we will discuss in the next paragraph. Hoarfrost can have variable amounts depending on the amount of available moisture (see also Section 5.2). However, as can be seen in Fig. 2, hoarfrost mainly occurs in winter, at deep temperatures when absolute humidity is comparatively low. While 130 days with hoar frost have temperatures between -60 and -70C, about 70 hoar frost days are in the temperature range -60C to -50C; only less
than 30 days show temperatures higher than -50C. This means that hoar frost does not have a specific fingerprint due to its crystal type and formation process, as stated in a preliminary study by Stenni et al. (2016), but, as speculated about qualitatively by Stenni et al. (2016) already, the different signal is due to the low temperatures prevailing at days with hoar frost formation.

Note that Fig. 2 only displays the number of days with the observed precipitation type and does  not take into account snowfall amounts. Snowfall days at higher temperatures are less frequent than those at temperatures below -50 °C, but usually have considerably larger amounts of precipitation.

Figure 3 displays the wind direction at the AWS Dome C II for a) all days and b) only observations with wind speeds above 10 ms$^{-1}$. Dome C is the Antarctic location with the highest constancy in wind direction (Wendler and Kodama, 1984), even though no katabatic influence is found at the dome. Wind directions still show a preference for the SW sector, which can be explained by the climatological mean pressure distribution with an anticyclone prevailing above the continent that, on average, leads to approximately westerly to southerly winds at Dome C. For the higher wind speeds, the direction is much more variable, which demonstrates that the prevailing anticyclonic weather conditions are disturbed more often than previously thought.

**4.2 Synoptic patterns during precipitation**

Based on mainly 500 hPa geopotential height from the AMPS archive, six different synoptic situations that lead to increased amounts of precipitation were classified. The classification was done manually because it allows the investigator to be in full control and oversight of the process and because the classification system can be tailored precisely to the researcher's needs. Figure 4 displays examples for these six classes:

4a) Blocking anticyclone

Figure 4a shows the 500 hPa geopotential height field for 23 May 2007 00 UTC. A strong upper-level ridge is situated between 130 °E and 160 °W, with the corresponding trough west of it and the ridge axis extending from NNW to SSE, which consequently brings Dome C into a strong northwesterly flow that originates at a latitude of approximately 45 °S. The relatively warm and moist air from these latitudes is orographically lifted above the Antarctic continent, which leads to cooling and precipitation formation. Even though only a small fraction of the original moisture arrives at Dome C , it is enough to produce precipitation amounts about one order of magnitude larger than the more frequent diamond dust precipitation. The pattern lasted from 22 to 26 May 2007 in almost the same configuration, thus leading to a considerable amount of precipitation. Note that "considerable" on the high East Antarctic plateau, where annual precipitation is in the order of 20- 30 mm water equivalent (w.e.), means 24h – precipitation amounts of 0.1-1.0 mm w.e. However, this synoptic precipitation is generally one order of magnitude higher than diamond dust precipitation, which usually exhibits values clearly below 0.1 mm.w.e. The AMPS accumulated precipitation (12h-36h

forecast from 22 May 12 UTC corresponding to the precipitation total for the period 23 May 00 UTC-24 UTC) is also shown. It can be clearly seen how precipitation decreases from the coast towards the interior, but still reaches the high plateau.

4b) Weak anticyclone with north-westerly flow

Figure 4b displays similar fields as in Fig. 4a, the 500 hPa geopotential for 13 Feb 2007 00 UTC and the 24 h precipitation for 13 Feb. The high pressure ridge, situated slightly farther to the west than in the previous case, is of smaller meridional extent than in the Fig. 4a and is less persistent, but principally the situation is fairly similar, with transport of moist, warm air in a northwesterly flow between an upper level ridge and a trough from areas south of 50 °S. Those situations occur fairly frequently (order of magnitude: once per month, although with high inter-annual variability).

4c) Anticyclone with north-easterly flow

In Figure 4c a special case of the earlier examples is shown: specifically the flow here is northeasterly rather than northwesterly. In this synoptic pattern, often a cut-off low or upper level low is situated north or slightly northwest of the coast of Wilkes Land . The flow is directed around the cut-off low towards Dome C. While the distance to the coast is similar for a north-westerly and a northeasterly flow, some dynamic lifting of the air mass above the ocean might be involved in addition to the orographic lifting. This should be studied in a future investigation.

4d) Splitting of  flow

In contrast to conditions determined from studies for Dome Fuji and Kohnen Station in Dronning Maud Land, Dome C relatively often experiences a situation where the planetary waves are amplified, but the flow is split into a zonal part, in which Dome C is situated, and a meandering part with the strong trough and ridge in the amplified flow staying north of the Dome C region. While this leads to reduced advection of warm and moist air to Dome C, it can still cause precipitation formation. As the air mass originates farther south than in the cases described above,  the meridional exchange of heat and moisture is smaller.

4e) Flow from West Antarctica

Another situation that has not been found at other deep drilling sites is that relatively warm
and moist air is advected to Dome C from the Amundsen-Bellingshausen Sea across Marie
Byrd Land. In the 500 hPa geopotential height field in Figure 5e) a closed circulation centered
in the Ross Sea can be seen, which leaves Dome C in a flow transporting air from the
Amundsen Sea or north of it towards Dome C. For this situation, AMPS shows precipitation
only in the vicinity of Dome C, not at the station itself. This situation is most likely influenced
by the strength and position of the Amundsen-Bellingshausen Seas Low (ASL, e.g. Raphael et
al., 2016). Fig. 5 shows the AMPS sea level pressure for 3 May 2007. In the described case,
the ASL is found in a rather western position, corresponding to its usual annual cycle, which
features a westernmost position in winter. A small but strong core is found in the western
Amundsen Sea, accompanied by an upper level low. Together with a weaker but broader low
at surface and upper levels above the Ross Sea and beyond, this leads to a northerly flow at
the eastern edge of the ASL, which continues over the continent towards Dome C.

4f) Post-event increased moisture

Several cases, for which AMPS shows very low or no precipitation, exhibit increased amounts
of measured precipitation at Dome C. The precipitation was classified as diamond dust, but
the events showed amounts that were atypically high for this type of precipitation. It was
found that these cases, which did not show the northerly flow connected to advection of
relatively warm and moist air, usually occurred after a synoptic snowfall event had happened.
This implies that the available moisture was still increased, and AMPS shows a fairly large,
isolated area of weak precipitation almost centered at Dome C.

**4.3 Wind speed and precipitation**

The AMPS wind direction for synoptic precipitation events only, identified in the AMPS data,
is displayed in Fig. 6. Contrary to the average conditions displayed in Fig. 3, which have a
pronounced preference for the southeast sector, for snowfall events the most frequent wind
direction is NNW to NW, with almost no cases displaying flow from the SW sector. Also, the
highest wind speeds (12-14 ms$^{-1}$) are observed to come from a northwesterly direction.  Here
AMPS data rather than AWS data are used for this figure because the AMPS data were

utilized to identify the high-precipitation events. In the observations, some cases with high precipitation were not found because they were accompanied by high wind speeds, and thus no sampling of the precipitation was possible after the snow had been blown off the measuring platform. Comparison of the total precipitation amount derived from the sampling

to data from an accumulation stake array shows that the amount of sampled precipitation is lower than the measured accumulation. A study of the mismatches of AMPS and observation (i.e. where AMPS showed large precipitation amounts when no precipitation was reported in the observations) revealed that those cases usually showed an increase in temperature and wind speed observed at the AWS, indicating a synoptic disturbance. The annual number of

such events varies between 3 in 2010 and 10 in 2008; in 2009, 8 events were identified in the AMPS data. The fact that those dates are not included in our analysis most likely does not weaken our results; on the contrary, since they are all related to synoptic disturbances, they would appear to rather emphasize our findings.

This wind influence also becomes clear from Figure 7, in which the relationship between

precipitation amounts and wind speed is illustrated. Precipitation amounts are related to wind speed for a) observations and b) AMPS archive data.  Again, it has to be considered that days with high wind are mostly related to synoptic snowfall events that have high precipitation amounts in AMPS, but cannot be seen in the observation since the snow has been blown off the measuring platform and thus not been recorded. Thus, Fig. 7b seems to be more realistic

than Fig. 7a, with larger precipitation amounts at correspondingly higher wind speeds. Surface mass balance data from firn cores and a stake array suggest that AMPS precipitation has a positive bias, whereas the total amounts measured at the platform are too low, which seems plausible considering the above mentioned mass losses due to removal of snow from the platform by the wind. Since all three methods have considerable error possibilities, we refrain

from a more specific numeric quantification of these findings.

**4.4 Isotope measurements and modelling**

Figure 8a shows observed $\delta^{18}O$ vs. 2 m air temperature for the different types of precipitation:

snow, diamond dust, and hoar frost. High-precipitation events, for which trajectories were calculated, are marked with circles. The regression lines differ only slightly for the  various precipitation types. For all samples, a $\delta^{18}O$-T slope of 0.49 ‰/°C is found (r=0.79, n=498). The slope for the studied high-precipitation events only is  0.39 ‰/°C, lower than for all days (r=0.78, n=21). Also, the relationship between deuterium excess and $\delta^{18}O$ (Fig. 8b) shows no

significant differences between the precipitation types. Slopes for the different crystal types are discussed in detail and compared to other Antarctic sites in Stenni et al., (2016) (modelled / observed values, daily / monthly values, inversion/2m-temperature). At the time of our study, the time series of analysed samples was not long enough to calculate statistically significant slopes for the different synoptic situations. Since the Dome C precipitation series is being continued this should be possible in the future. The only precipitation – stable isotope data series in Antarctica that is sufficiently long to get statistically significant results is that of Neumayer Station, coastal DML (Schlosser et al., 2004). This data series has meanwhile been extended to 36 yr and is being re-investigated.

In Figure 9 the observed and modelled $\delta^{18}$O and deuterium excess for days with moisture source estimates are displayed. Observed $\delta^{18}$O and deuterium excess show a clear annual cycle. While $\delta^{18}$O exhibits a clear maximum in summer, the deuterium excess peaks in winter, most clearly in 2010, which was least disturbed by warm air intrusions. For the modelling of isotopic fractionation with MCIM, initial conditions at the moisture sources derived from a 5 day back-trajectory calculation combined with the synoptic flow analysis were used following the method described in Dittmann et al. (2016). Trajectories were calculated for all cases where the synoptic situation seemed to be suitable for it, meaning a rather clear atmospheric flow pattern. When this was not the case, trajectories tended to have kinks and loops and were not plausible or were suspect, and thus not included in the study.

The moisture sources for arrival levels 600 hPa and 500 hPa are shown in Figure 10. Stronger colours correspond to higher frequency of occurrence of the respective moisture source. For cases, in which the trajectory left the AMPS domain, ECMWF interim-reanalysis data were used to estimate the moisture source. For both arrival levels, the moisture source is found mainly in the 90 °E - 130 °E longitude range of the Southern (Indian) Ocean. The most frequent latitude ranges are 40 °S - 50 °S for 600 hPa arrival level, and 35 °S - 50 °S for the 500 hPa level. The model was run using different assumptions for the condensation temperature: i) moisture source conditions derived for the estimate using the 500 hPa back-trajectory: arrival temperature of the trajectory at the 500 hPa level (blue circles in Fig. 9), ii) moisture source conditions derived for the estimate using the 600 hPa back-trajectory; arrival temperature at the 600 hPa level (red circles), and iii) moisture source estimated using the 500 hPa trajectory; arrival temperature at the upper limit of the inversion layer derived from radiosonde data (green circles). For all model runs, the model parameters were kept constant. These parameters had been adapted to increase the calculated fractionation on the moisture

transport path in order to get the best agreement of modelled and observed values. The slope of the supersaturation over ice was set to zero and the parameters determining the amount of precipitation leaving the cloud were set close to Raleigh conditions. The modelled $\delta^{18}O$ values are generally too high, no matter which assumption is made for the condensation temperature. Using the 500 hPa data yields a smaller bias, but a lower correlation between the observed and modelled $\delta^{18}O$ than using the 600 hPa temperatures and moisture source assumptions. (R=0.61, bias= 3‰ and R=0.74, bias=11.3 ‰ for 500 hPa and 600 hPa, respectively, p<0.05). The corresponding values for use of the inversion temperature are R=0.66 (p<0.05), bias = 10.5 ‰. An attempt to use the condensation temperature at Dome C derived from radiosonde data as model input (rather than inversion temperature or temperature at the arrival levels of the calculated trajectories) did not improve the correlation between observed and modelled isotope ratios: no statistically significant correlation between modelled and observed $\delta^{18}O$ was found in this case. The correlation of condensation temperature $T_C$ and 2m- temperature $T_{2m}$ was clearly weaker than the correlation of, $T_{inv}$, $T_{500}$ and $T_{600}$ with $T_{2m}$. Since $T_{2m}$ and observed $\delta^{18}O$ were well correlated, it could not be expected that using $T_C$ in the model would yield better results for the stable isotope ratios. $T_c$ also showed a weaker annual cycle than Tinv. Modelled and observed deuterium excess show a weak correlation only when the inversion temperature is used as condensation temperature (R=0.51, p=0.02). Ekaykin et al. (2009) investigated the relationship between the inversion temperature and the 2m temperature and their correlation with stable isotope ratios for longer time scales. They state that in central Antarctica the condensation temperature is considerably lower than the temperature at the top of the inversion layer, since diamond dust forms throughout the inversion layer. However, this should not matter in our case because for the modelling we considered cases of synoptic snowfall rather than diamond dust formation.

It should be kept in mind, though, that MCIM is a relatively simple model with various strong simplifications, such as assumptions of a single moisture source, a single temperature inversion, and a humidity inversion parallel to the temperature inversion. Additionally, it is assumed that the 500hPa level is representative for the general moisture transport, which is likely, but maybe not true in all cases. Also, while the moisture source is estimated as exactly as possible, there might be errors here that could also lead to a weaker agreement of modelled and observed stable isotope ratios. Lastly, the determination of the lifting condensation level using radiosonde data in the extremely dry atmosphere above Dome C is a challenge and may also introduce errors.

## 5 Comparison of Dome C and Dome Fuji

Dome C and Dome Fuji are both deep ice-core drilling sites with the oldest ice ever drilled on earth (800.000yr, and 720.000 yr, respectively, (EPICA community members, 2004: Motoyama, 2007)). At 3810m altitude and 77.31 S, 39.70E, Dome Fuji  is situated slightly further south and at an approximately 600m higher elevation than Dome C. Consequently, Dome Fuji has a slightly lower mean annual temperature of -57.7C (compared to -54.5C at

Dome C). Accumulation rates are very similar (25mm for Dome C, 27mm for Dome Fuji). Generally both locations experience the same extremely dry and cold climate of the high East Antarctic Plateau. Being on top of a topographic dome, neither one is under the influence of katabatic flow.

Whereas daily precipitation measurements are available at Dome Fuji for only one year, the

Dome C series is a multi-year time series having been continued to the present. For our study, the years 2008-2010 were analysed. In addition to the type of data used in the Dome Fuji study, at Dome C upper-air data and crystal type data were available for our study. The synoptic situations responsible for precipitation are fairly similar for both stations (basically related to amplified Rossby waves); however, cases specific to either Dome C or Dome Fuji

do occur: Whereas for Dome Fuji this refers to a situation with moisture advection from the south (via Kohnen Station (Schlosser et al., 2010a: 2010b), another deep drilling site), for Dome C the moisture is advected via West Antarctica and the flow related to the ASL. The latter is of special importance for glacial periods when the topography of the ice sheet was different from today.

The case of splitting of the flow (Fig. 4d) did not occur in the Dome Fuji study, however, given the shortness of the investigation period, we cannot rule out the possibility that this situation also occurs in the Dome Fuji area.

Our study generally confirms the results of the Dome Fuji study. Both studies pointed out that the synoptic situation of amplified waves with strongly developed troughs and ridges lead to

increased meridional exchange of heat and moisture. For interpretation of stable isotope profiles from ice cores, this means that a more northern moisture source does not necessarily mean a stronger depletion in heavy isotopes since the temperature difference between moisture source and deposition site is reduced. Also, based on daily values or precipitation events, no correlation between deuterium excess and moisture source conditions could be

found for either location. Most earlier studies deal with longer time periods (from at least months-seasonal  to glacial/interglacial changes).

## 6 Discussion and Conclusion


The first and only multi-year data series of daily precipitation amounts, precipitation type and stable isotope ratios at an Antarctic deep ice core drilling site was combined with output from a mesoscale atmospheric model and a simple isotope model to study the influence of the precipitation regime on the corresponding stable water isotope ratios.

Here we present the first complete classification of synoptic patterns for precipitation events at Dome C for 2008-2010. Snowfall events with precipitation amounts an order of magnitude larger than diamond dust precipitation were often associated with amplification of Rossby waves in the circumpolar trough with increased meridional transport of heat and moisture.

In contrast to other deep drilling sites in East Antarctica, such as Dome Fuji (Dittmann et al.,
2016) and Kohnen (Schlosser et al., 2010), at Dome C in some cases a moisture transport from West Antarctica across the continent occurred. This is particularly interesting due to its relation to the Amundsen-Bellingshausen Sea Low (ASL). Strength and location of the ASL have a strong influence on meridional exchange of heat and moisture in West Antarctica (Raphael et al., 2016).


The $\delta^{18}$O-T relationship did not differ considerably between the different precipitation types: snowfall, diamond dust and hoarfrost showed almost similar slopes. Hoarfrost exhibited significantly lower $\delta^{18}$O and $\delta$D values and higher deuterium excess than snowfall and diamond dust. Whereas Stenni et al. (2016) state that hoar frost has a distinct fingerprint
among the various precipitation types, implying that moisture sources and or the hydrological cycle might be different for hoarfrost, our current, more detailed study has shown that this "fingerprint" is due to the fact that hoarfrost occurs predominantly during the cold period. Relatively large amounts of hoarfrost are measured after synoptic snowfall events, when humidity is still increased after moisture transport from lower latitudes, implying that
hoarfrost basically has the same moisture sources as the other precipitation types. The local cycle of sublimation and deposition of hoar frost is still fairly unknown, but seems to be a process where the depletion and enrichment of heavier isotopes are reversible. This leads to the conclusion that since there is no moisture source on the continent, the moisture responsible for diamond dust and hoar frost formation has to be transported on similar
pathways as synoptic snowfall to the interior of the continent.

Note that diamond dust is not parameterized in the WRF model used in AMPS. Nevertheless the model output used here yields only 6 days with no precipitation at all in the study period. Modelled stable isotope ratios showed a "warm" bias compared to the observations, which was also found in previous similar studies (e.g. Steen-Larsen et al., 2017).

However, using the condensation temperature at Dome C derived from radiosonde data as model input (rather than the temperature at the top of the inversion layer or the temperature at the arrival levels of the calculated trajectories) did not improve the correlation between observed and modelled isotope ratios; in fact, the correlation coefficient decreased considerably and was no longer significant, most likely because the condensation temperature

determined from the radiosonde data displayed only a weak annual cycle. More detailed studies of vertical humidity and temperature profiles during precipitation are necessary to understand this result. However, at present, no explanation for this can be offered. The assumption generally used in ice core studies (e.g. Stenni et al. 2016) that the temperature at the top of the inversion layer represents the condensation temperature could not be proven.

No correlation was found between observed deuterium excess and relative humidity at the estimated moisture source, which is contradictory to measurements by Uemura et al (2008) and Steen-Larsen et al. (2014). Whether this has general physical reasons or is due to the fact that we studied individual events or due to errors in moisture source estimation, cannot be determined with the given data set.

It was also found that a more northern moisture source does not – as commonly assumed - necessarily mean stronger depletion of heavy isotopes, since the advection of warm air associated with snowfall events reduces the temperature difference between oceanic moisture source and deposition site, and thus reduces the strength of the distillation. This confirms the recent results of Dittmann et al. (2016) found at the deep drilling site Dome Fuji for a 1-yr

time period.

With the extension of the data series in the future it will be possible to calculate statistically significant delta-T slopes for the different synoptic situations. Combined with simulations of the past climate with General Circulation Models (GCMs) this will lead to a more exact, quantitative interpretation of stable isotope profiles from deep ice cores. However, more

multi-year precipitation data sets are needed in Antarctica for a better spatial representativeness.


## Author contribution:

Barbara Stenni is responsible for the precipitation measurements and stable isotope analysis, Mauro Valt and Anselmo Cagnati for the crystal analysis, and Paolo Grigioni and Claudio

Scarchilli for the radiosonde data provision and analysis. Anna Dittmann carried out the stable isotope modelling, with contributions by Valerie Masson-Delmotte, as well as the comparisons of observations with modelled meteorological and isotope data. Elisabeth Schlosser did the analysis of synoptic patterns, where AMPS data analysis was supported by Jordan Powers and Kevin Manning. The manuscript was prepared by Elisabeth Schlosser,

Anna Dittmann, Jordan Powers, and Kevin Manning with constructive comments of the other co-authors.

## Acknowledgements

This study was funded by the Austrian Science Funds (FWF) under grants P24223 and P28695. AMPS is supported by the U.S. National Science Foundation, Division of Polar Programs. The precipitation measurements at Dome C have been carried out in the framework of the Concordia station and ESF PolarCLIMATE HOLOCLIP projects. We appreciate the support of the University of Wisconsin-Madison Automatic Weather Station Program with the

Dome C II data set (NSF grant numbers ANT-0944018 and ANT-12456663). Radiosonde data and information were obtained from IPEV/PNRA Project "Routine Meteorological Observation at Station Concordia – www.climantartide.it. We would like to express our gratitude to all winterers at Dome C, who were involved in the precipitation sampling.

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

**Figure Captions**

**Figure 1**

Histogram of daily precipitation amounts for a) measurements and b) AMPS archive data for the period 2008-2010.

**Figure 2**

Frequency of the different precipitation types of observed precipitation

**Figure 3**

a) Observed wind speed $W_s$ and direction at Dome C AWS

b) Observed wind speed $W_s$ and direction for cases with wind speeds larger than 10m s$^{-1}$

**Figure 4**

Synoptic patterns classification:

500 hPa geopotential height (contour interval 10gpm) and 24h precipitation totals from
930 AMPS archive for the different synoptic situations during precipitation:

a) blocking anticyclone with northwesterly flow (23 May 2007)

b) weak anticyclone with northwesterly flow (13 Feb 2007)

c) anticyclone with northeasterly flow (16 Mar 2007)

d) splitting of flow (14 Aug 2008)

e) southerly flow from West Antarctica (3 May 2007)

f) post event (28 May 2007)

500 hPa geopotential height fields stem from AMPS 12h forecast of the preceding day,
corresponding to 00 UTC of the described day; precipitation is AMPS 12h-36h forecast of the
preceding day, corresponding to 00-24 UTC of the day described.

**Figure 5**

 Sea level pressure from AMPS (domain 1) for 3 May 2007 00 UTC

**Figure 6**

AMPS wind speed $W_s$ and direction for snowfall events identified in AMPS data

**Figure 7**

Observed wind speed at AWS vs. observed and modelled 24h precipitation at Dome C

**Figure 8**

a) $\delta^{18}$O of precipitation samples vs. 2m air temperature from AWS for the different types of precipitation, snow, diamond dust, and hoar frost. High-precipitation events, for which trajectories were calculated, are marked with circles.

b) $\delta^{18}$O of precipitation samples vs. deuterium excess

**Figure 9**

Observed and modelled a) $\delta^{18}$O and b) deuterium excess for days with moisture source estimates with Dome C temperature taken at 500hPa level, 600hPa level, and at the upper limit of the temperature inversion layer (derived from radiosondes as described in the text).

The green squares mark cases, for which trajectory calculations were carried out.

**Figure 10**

Estimated moisture source areas for arrival levels a) 600 hPa and b) 500 hPa. Stronger color corresponds to higher frequency of occurrence of the respective moisture source.





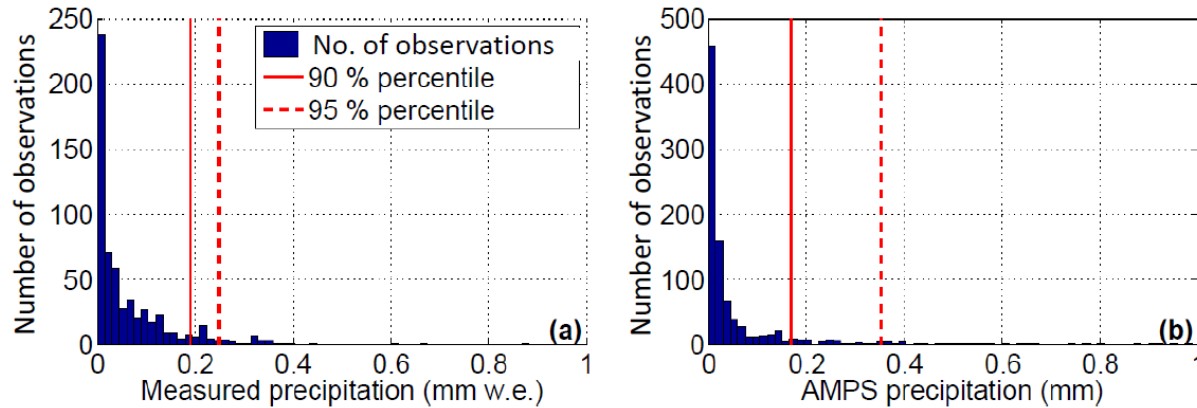

**Figure 1: Histogram of daily precipitation amounts for a) measurements and b) AMPS archive data for the period 2008-2010.**

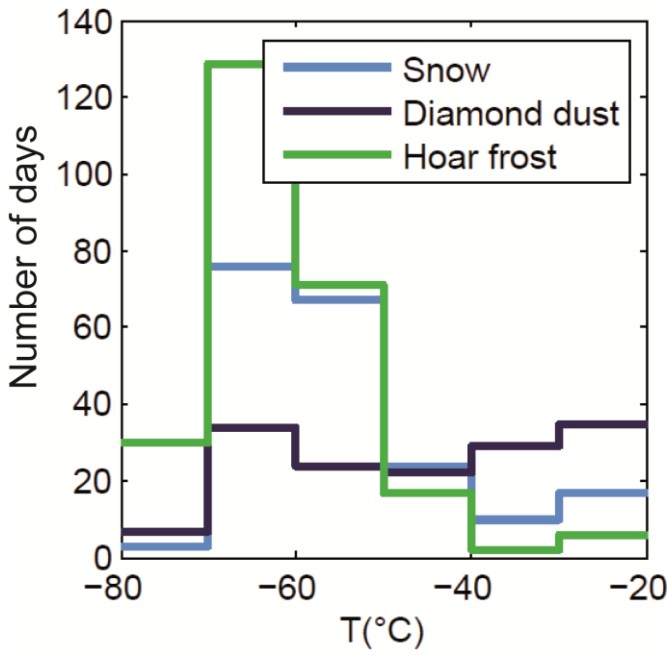


**Figure 2: Frequency of the different precipitation types of observed precipitation**

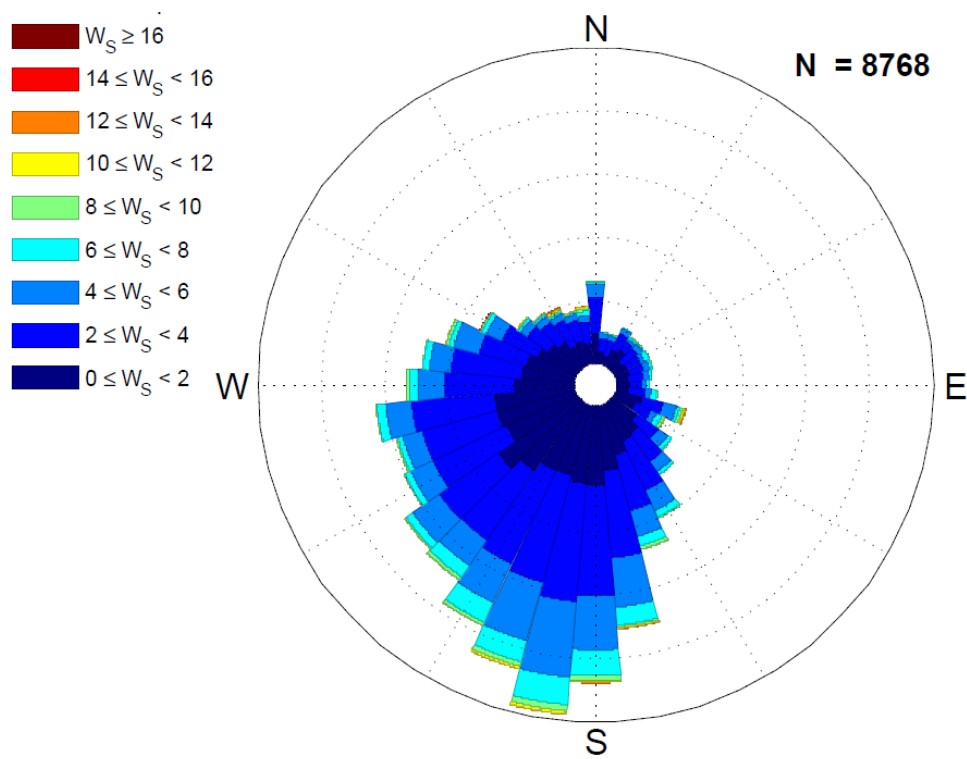

**Figure 3a): Observed wind speed W_s and direction at Dome C AWS**

$$\text{Figure 3a): Observed wind speed } W_s \text{ and direction at Dome C AWS}$$


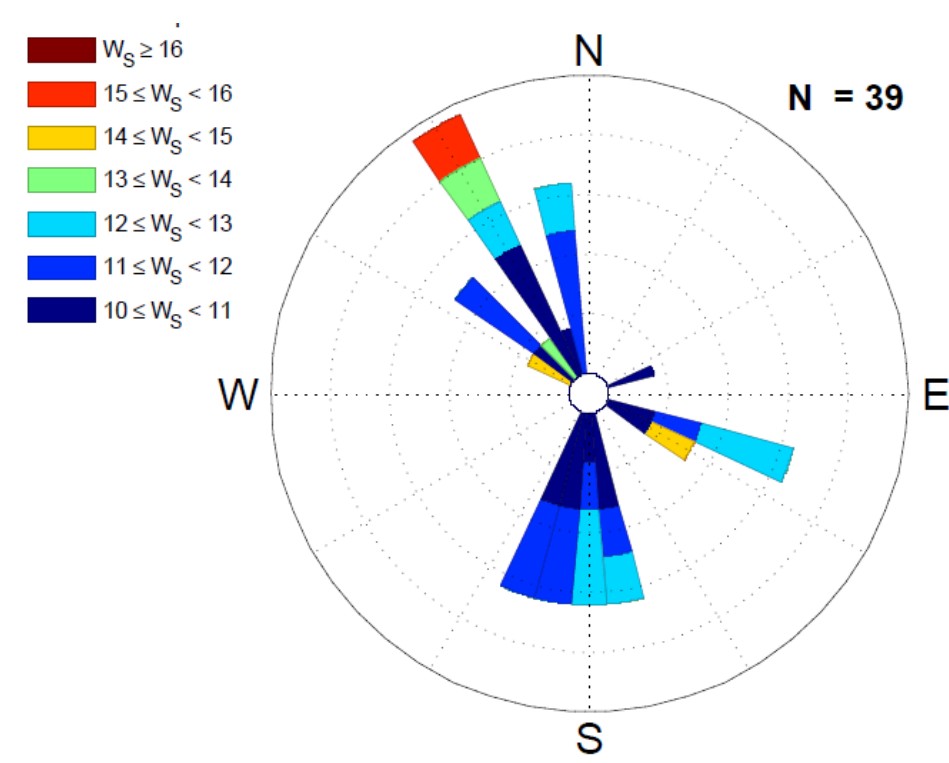

**Figure 3b): Observed wind speed $W_s$ and direction for cases with wind speeds larger than 10m s$^{-1}$**

**a)** blocking high 23 May 2007 00Z

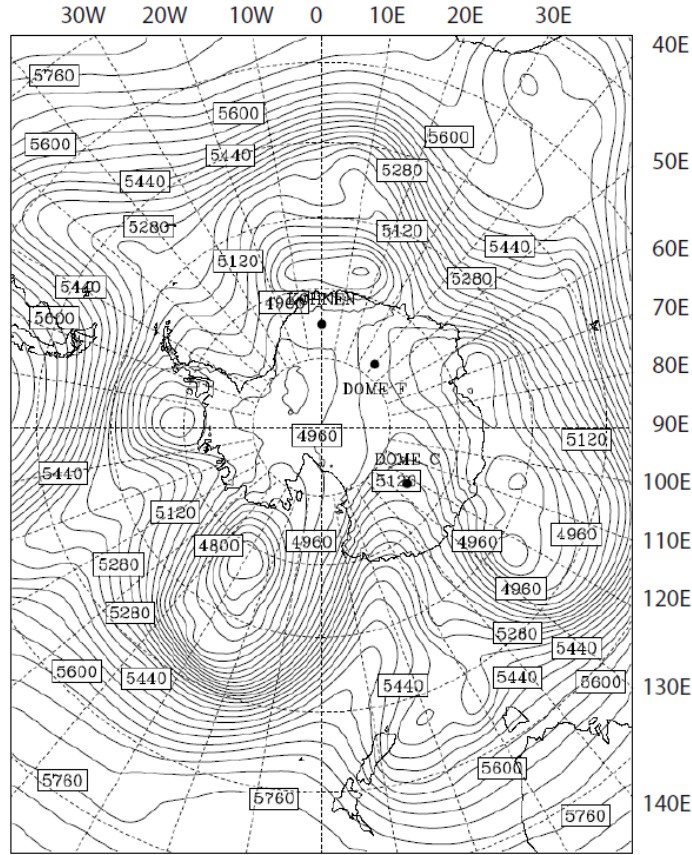


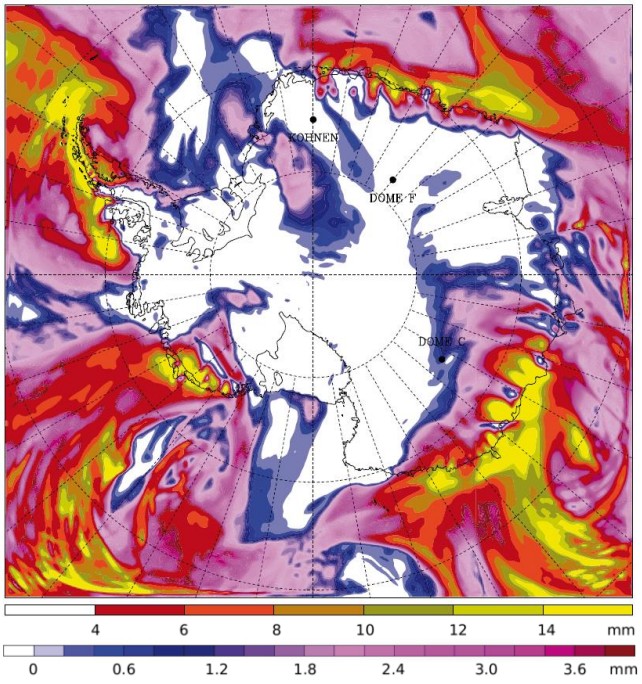

b) weak anticyclone with northwesterly flow

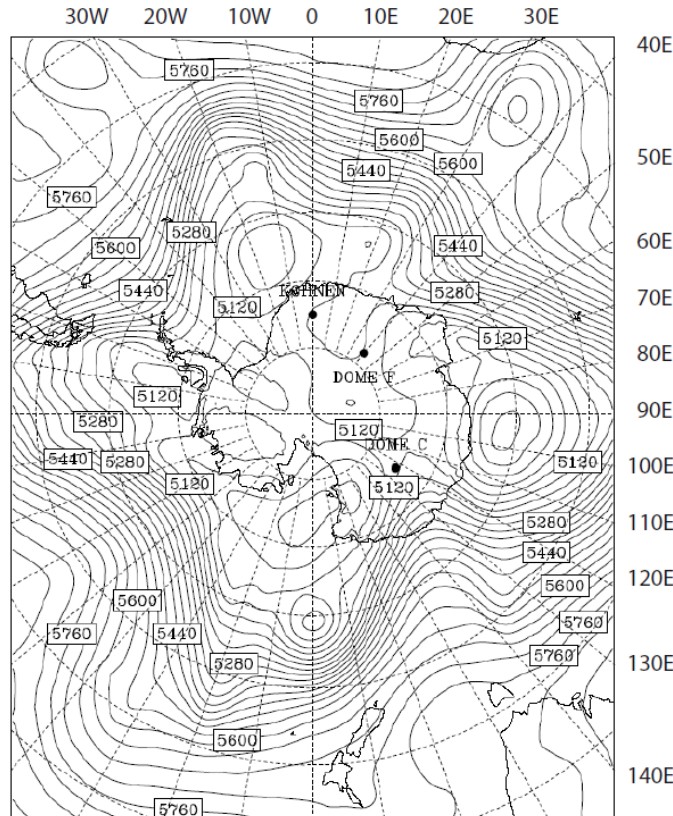


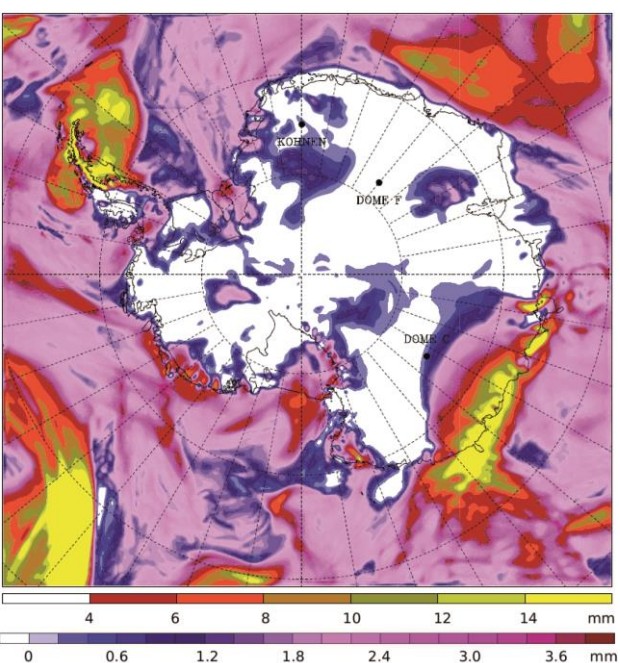

c) anticyclone with northeasterly flow 15 Mar 2007 00Z

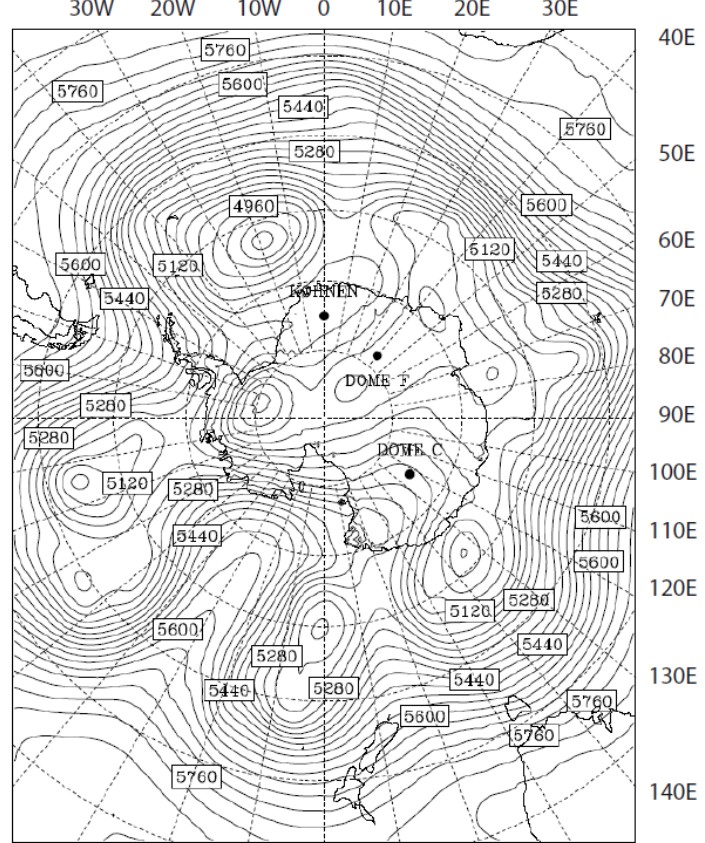

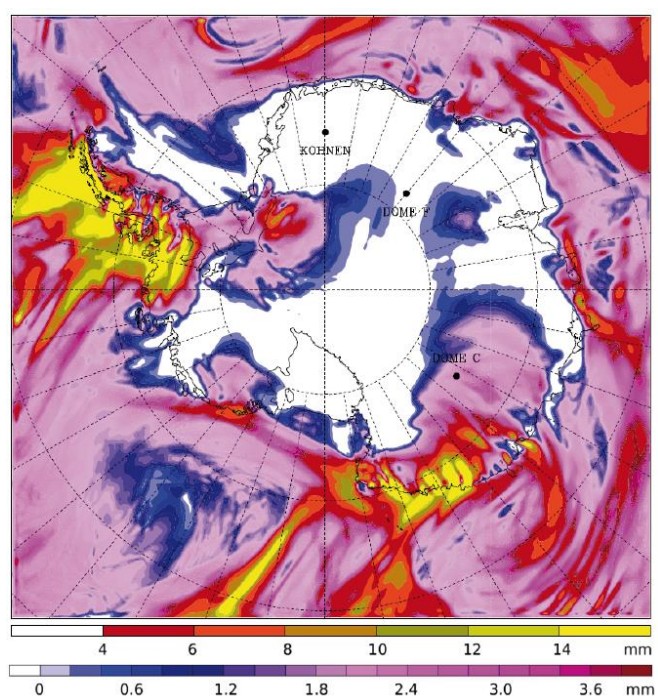

d) splitting of flow

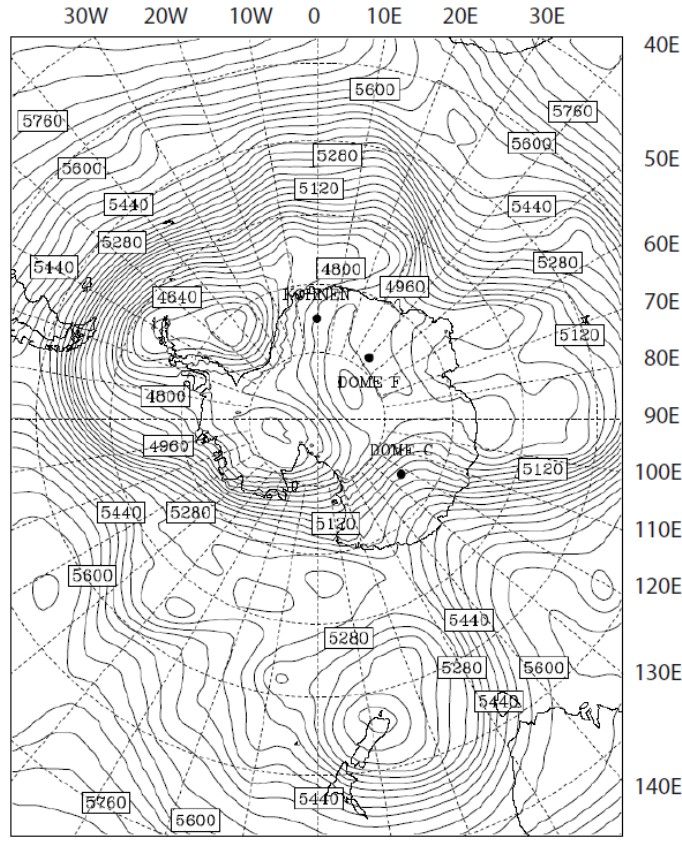

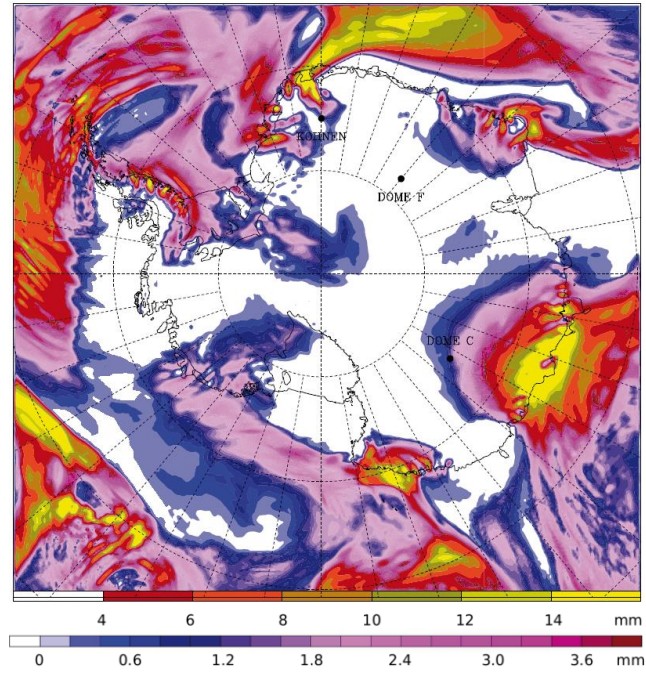


e) southerly flow from West Antarctica

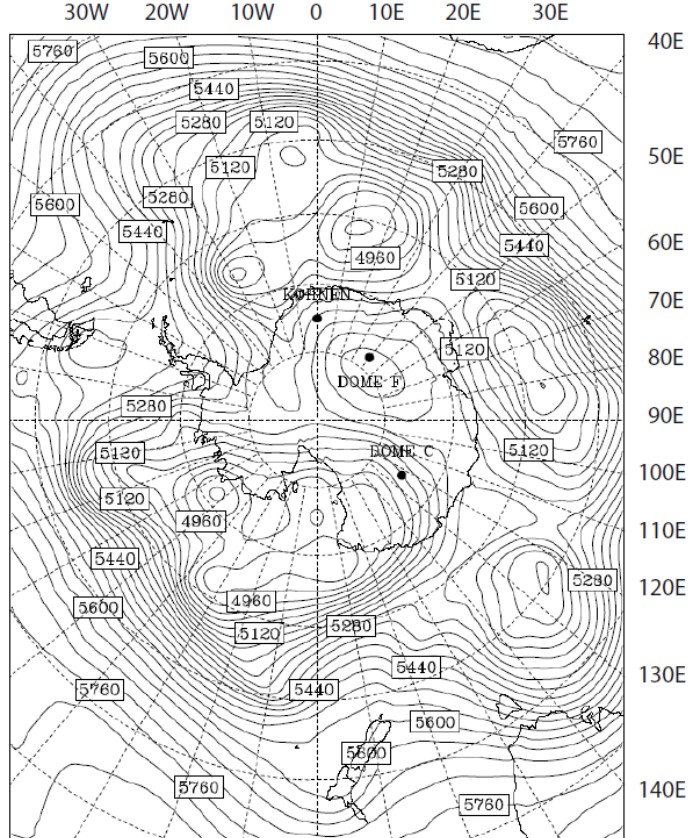

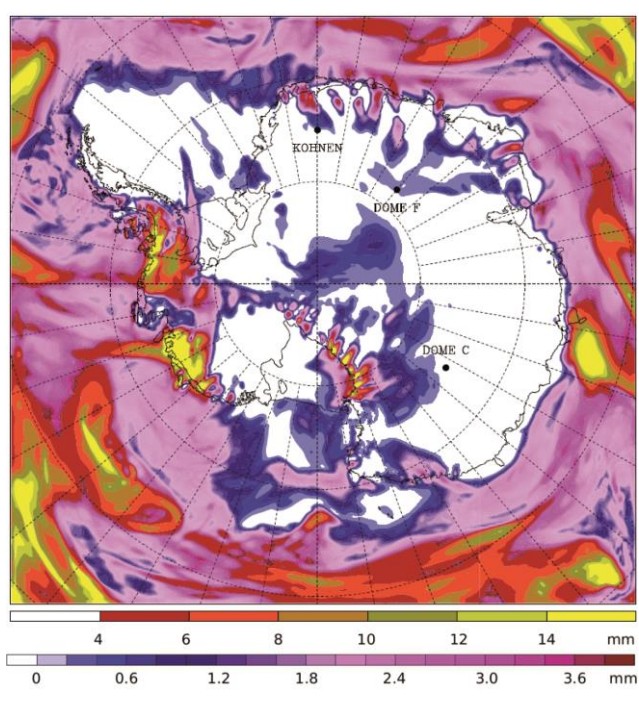


f) post event

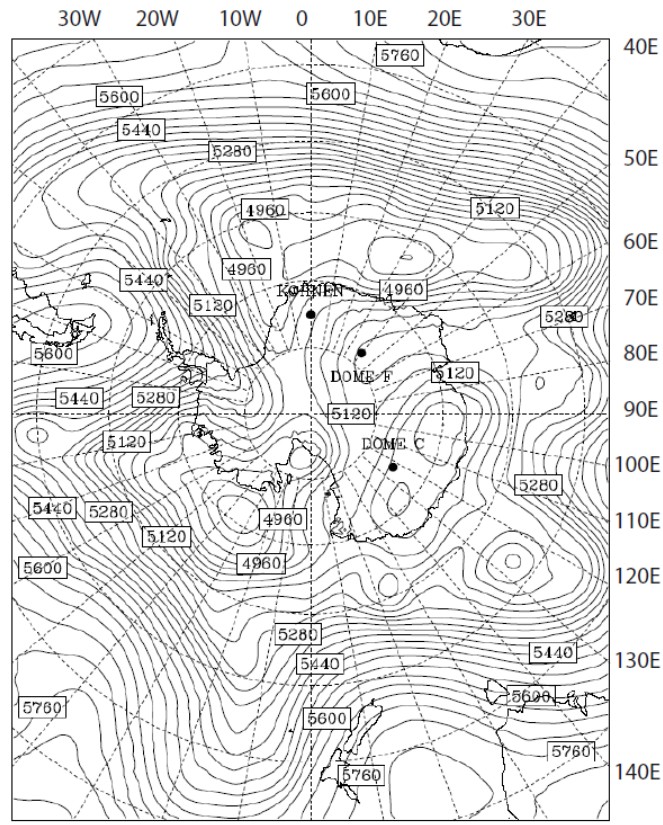

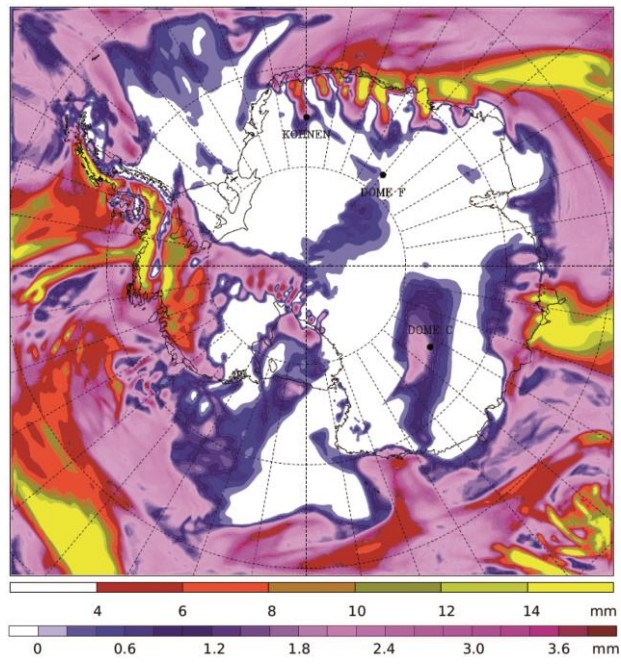


**Figure 4: Synoptic patterns classification**

500 hPa geopotential height (contour interval 10gpm) and 24h precipitation totals from AMPS archive for the different synoptic situations during precipitation:

a) blocking anticyclone with northwesterly flow (23 May 2007)

b) weak anticyclone with northwesterly flow (13 Feb 2007)

c) anticyclone with northeasterly flow (16 Mar 2007)

d) splitting of flow (14 Aug 2008)

e) southerly flow from West Antarctica (3 May 2007)

f) post event (28 May 2007)

500 hPa geopotential height fields stem from AMPS 12h forecast of the preceding day, corresponding to 00 UTC of the described day; precipitation is AMPS 12h-36h forecast of the preceding day, corresponding to 00-24 UTC of the day described.

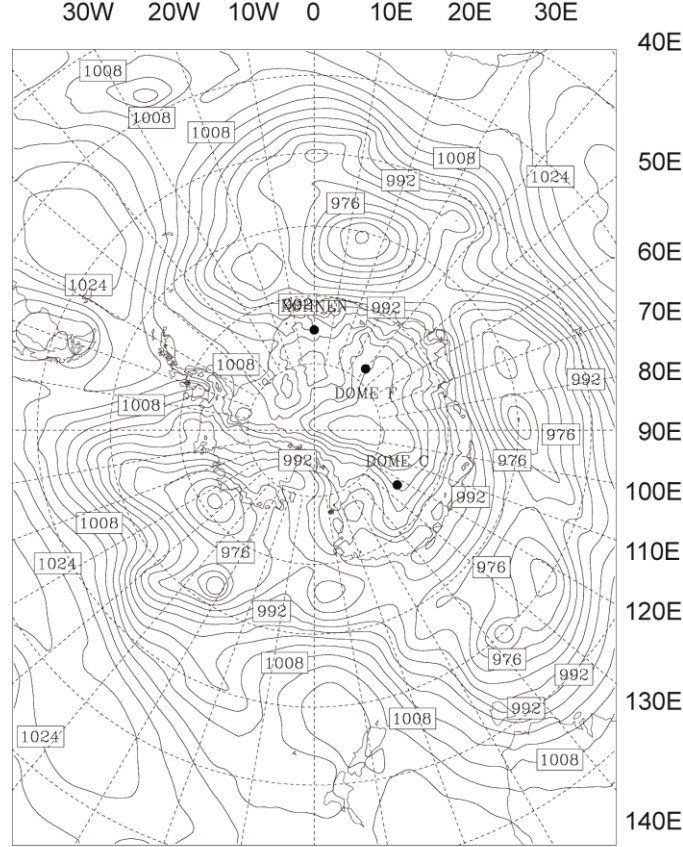

**Figure 5: Sea level pressure from AMPS (domain 1) for 3 May 2007 00 UTC**

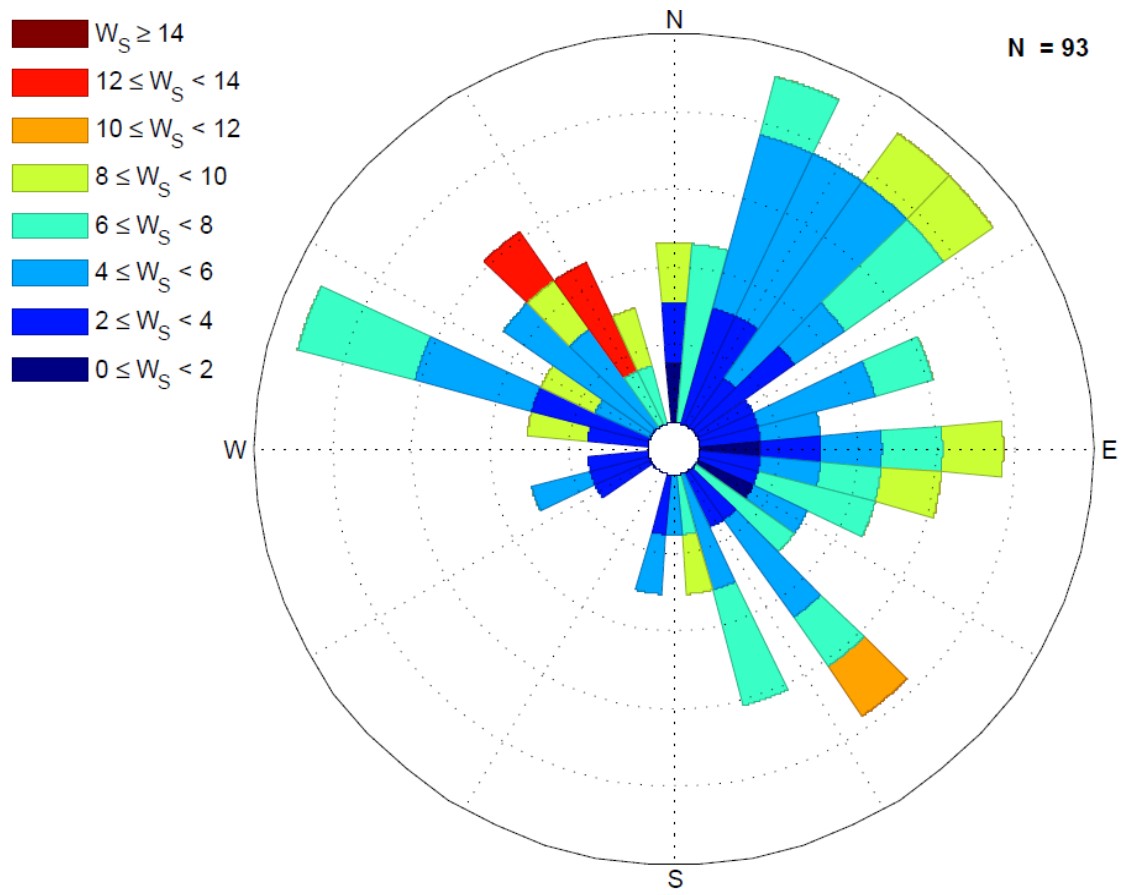

**Figure 6: AMPS wind speed $W_s$ and direction for snowfall events identified in AMPS data**

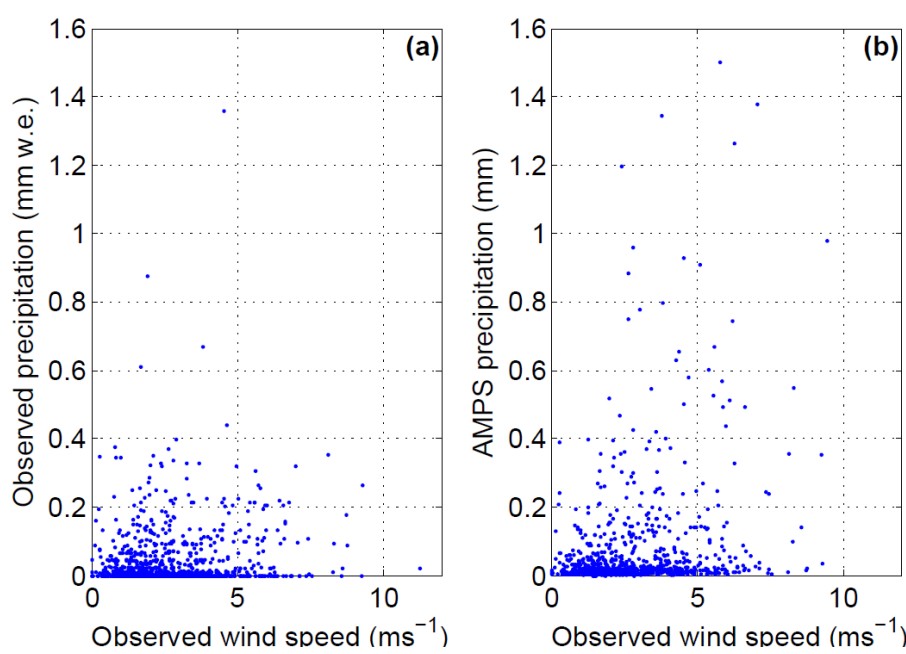

**Figure 7: Observed wind speed at AWS vs. observed and modelled 24h precipitation at**

 **Dome C**

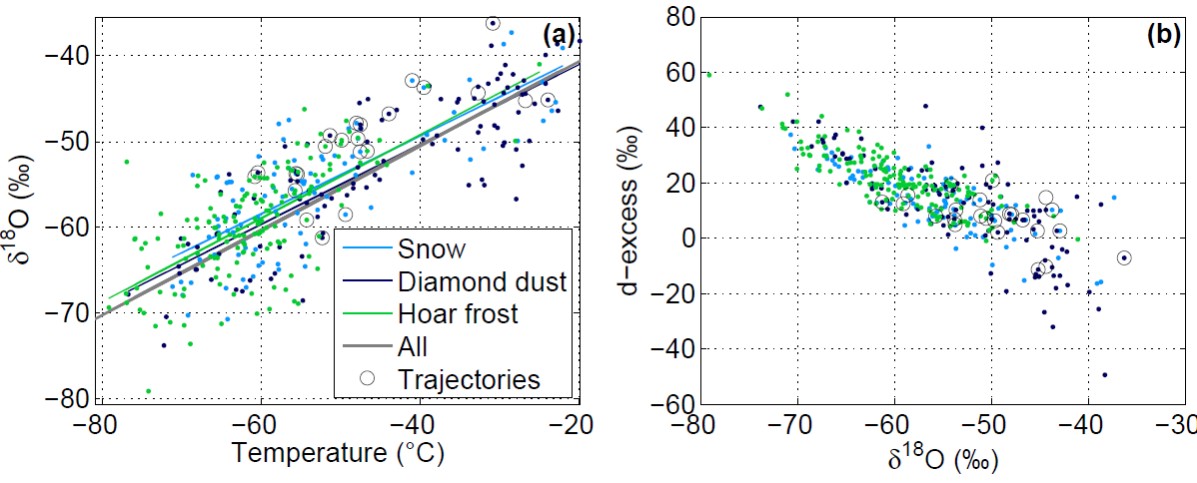

**Figure 8:**

a) $\delta^{18}O$ of precipitation samples vs. 2m air temperature from AWS for the different types of precipitation, snow, diamond dust, and hoar frost. High-precipitation events, for which trajectories were calculated, are marked with circles.

b) $\delta^{18}O$ of precipitation samples vs. deuterium excess

a)

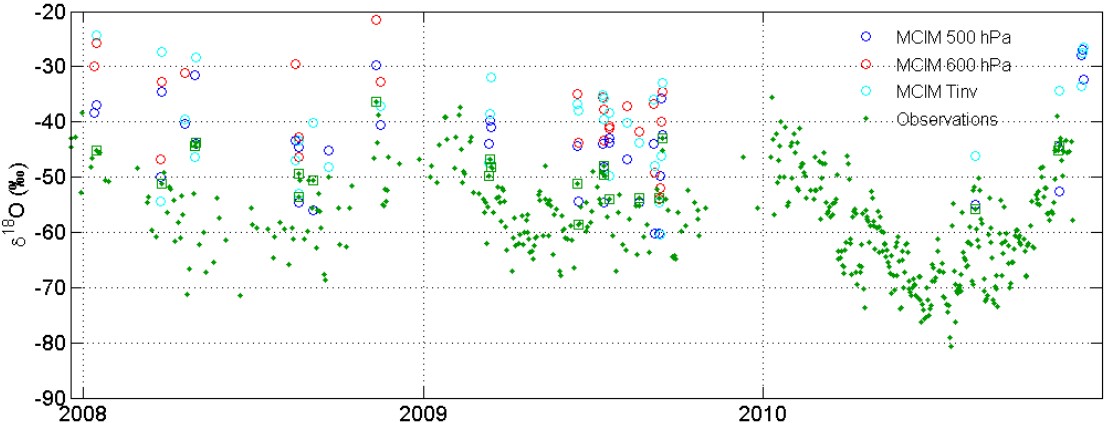

b)

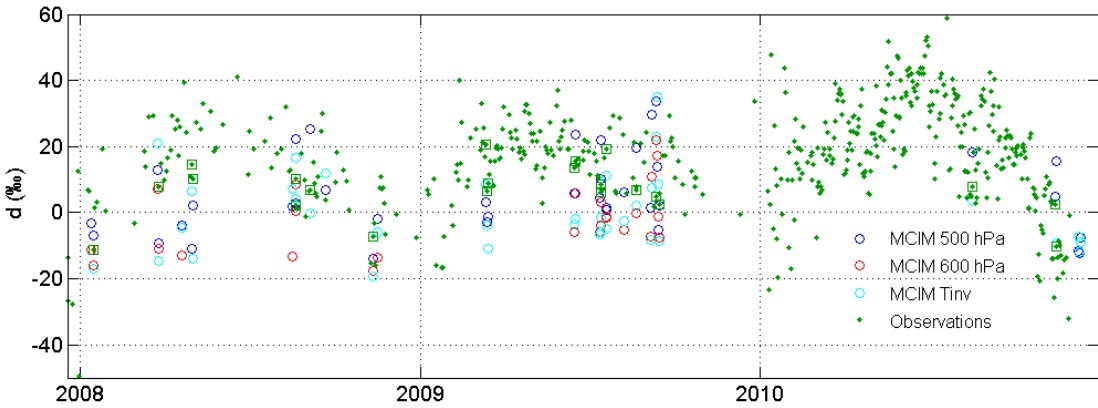

**Figure 9: Observed and modelled a) $\delta^{18}$O and b) deuterium excess for days with**
1085 **moisture source estimates with Dome C temperature taken at 500hPa level, 600hPa**
**level, and at the upper limit of the temperature inversion layer (derived from**
**radiosondes as described in the text). The green squares mark cases, for which**
**trajectory calculations were carried out.**

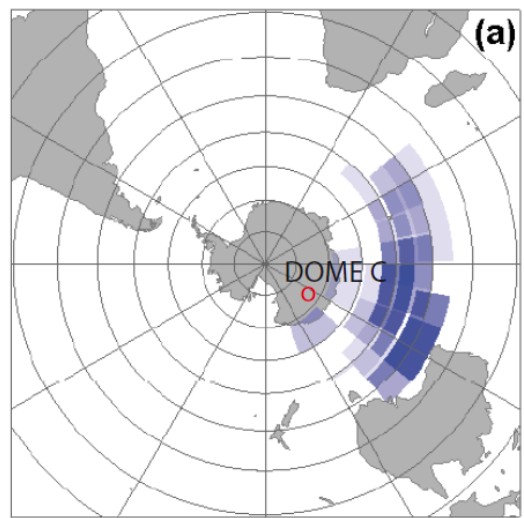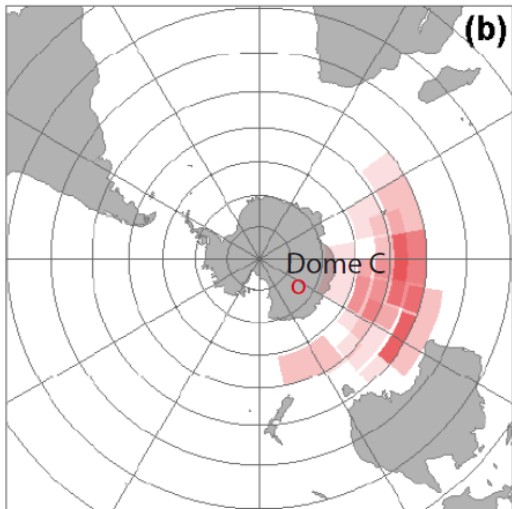

**Figure 10: Estimated moisture source areas for arrival levels a) 600 hPa and b) 500 hPa.**
 **Stronger colour corresponds to higher frequency of occurrence of the respective moisture source.**