# Peer review of "The influence of the synoptic regime on stable water isotopes in precipitation at Dome C, East Antarctica"

_The Cryosphere, 2017_

## Referee Comment (RC1) · Anonymous Referee #1 · 3 Mar 2017

This manuscript presents published measurements of d18O and d-excess of precipitation at the Dome C site as well as an application of a modeling approach combining the mesoscale atmospheric model and a simple isotopic model. The authors conclude that the model underestimates the depletion of d18O in precipitation in Antarctica.

This study does not provide any new data, despite the fact that the "Precipitation and stable isotopes data" are not presented in the part "Previous work". Everything has already been published in the paper by Stenni et al. (The Cryosphere, 2016).

The results on Dome C meteorological conditions and synoptic patterns during precipitation are already largely shown and discussed in the paper by Schlosser et al. (ACP, 2016).

The results of the isotopes modelling (part 5.4) have already been discussed largely by Dittmann et al. (ACP, 2016) for the Dome F site with the same conclusion. I thus do not see the added value of this study which is basically only a second application of the Dittmann et al. (2016) study on another site with similar characteristics.

The main conclusion of this paper and of Dittmann et al. (2016), i.e. that the MCIM does simulate too high d18O in Antarctic precipitation is not new. This has already been noted for example in Uemura et al. (CP, 2016).

I also feel that the introduction part is misleading with very few references to previous studies while much has been done in the recent years on the study of precipitation patterns and water isotopic composition in sites of the Antarctic plateau. The 2 recent papers mentioned in the introduction refer to Greenland studies. Similarly, the conclusion is very poor and only rephrase conclusions from previous studies (Schlosser et al., 2016; Dittmann et al., 2016; Stenni et al., 2016) without anything more.

I thus do not recommend publication of this manuscript which does not provide any scientific added value.
* * *

---

## Author Comment (AC1) · 6 Mar 2017

We appreciate Referee #1's effort to provide a review within short time. However, we got the impression that several things have been overlooked or understood incompletely.

Although it is true that a large part (not all!) of the data used in this study are already published, they have never been combined in the way we did it in the presented manuscript and were also supplemented by additional data. In order to make a publication self-contained, it is almost always necessary to explain some things that have been published before. We would like to stress that our study does yield new results that have not been published elsewhere.

[Figure]

Stenni et al. (2016) stresses the relationship of stable isotope ratios with meteorological station data. They discuss in detail the delta-T slope for various time periods and isotope variables, compare this to other locations and also look at the relationships amongst the isotope variables. The general atmospheric flow conditions are discussed only briefly, whereas in the new study we present a detailed analysis of the synoptic situations that lead to precipitation at Dome C.

Stenni et al. (2016) also state that hoar frost has a distinct fingerprint among the various precipitation types, implying that moisture sources and or the hydrological cycle might be different for hoar frost. Our more detailed study showed that this "fingerprint" is due to the fact that hoar frost occurs predominantly during the cold period. Relatively large amounts of hoar frost are measured after synoptic snowfall events, when humidity is still increased after moisture transport from lower latitudes, which means that hoar frost basically has the same moisture sources as the other precipitation types.

In Schlosser et al. (2016), the stable isotopes served mainly as motivation for the study. They only discussed the meteorological conditions in two extreme years, without any isotope modelling or specific discussion of the stable isotope data and without any general analysis of the synoptic conditions during precipitation events. For instance, the conditions shown in Fig. 4d and 4e did not occur in the analysis of 2009 and 2010. Especially the situation in Fig 4e is highly interesting due to its relation to the Amundsen-Bellingshausen Sea Low. Nothing comparable occurs at Dome Fuji, so it was not discussed in the study by Dittmann et al. (2016).

Dittmann et al. (2016) used a very short time series (less than 1 yr) from a different Antarctic location to study synoptic conditions and model stable isotopes. Even if we had done only the same for Dome C, it would be a valuable result to confirm Dittmann's findings with a longer time series from another location.

Additionally, we used radiosonde data (not available for Dome Fuji for Dittmann's study) to determine the temperature at the lifting condensation level (LCL). This temperature,
additionally to the temperature at the upper limit of the inversion layer, was used as input for the isotope model. It is a surprising result that this did not improve the model simulations. It is a very critical point for the relationship between temperature and stable isotope ratio, WHICH temperature is considered here. For many years, the temperature at the top of the inversion layer has been used, which is a strong simplification and more research is needed here.

Our main conclusion is not that the model underestimates the isotopic depletion. Modelling is only a part of our study. For the isotope part, it is considerably more important that the deuterium excess showed no relationship with relative humidity or wind speed at the estimated moisture source. This relationship has been a general assumption in isotope studies for decades.

Likewise, the assumption that a more northern moisture source automatically means stronger depletion was shown to be not true for single precipitation events and the involved physics suggest that this applies generally to Antarctic precipitation. This confirmed the results of the Dome Fuji study.

We agree that it would be worthwhile to include more studies from the Antarctic plateau in the section "previous work". (Unfortunately, reviewers rarely agree about the length of the "previous work" and "introduction" sections.) We mentioned the two Greenland studies because they used continuous measurements of water vapour stable isotopes. This kind of work has only recently started in Antarctica, but we will try to discuss some more references in a newer version of our manuscript.

(There is no publication by Uemura et al., CP 2016 (as suggested by the referee) to be found on the CP homepage. We are not aware of any study by Uemera et al. that investigates/models data from single precipitation events.)

---

## Referee Comment (RC2) · Anonymous Referee #2 · 25 Apr 2017

**Summary**

This manuscript presents an original and unprecedented data set about precipitation at dome C, East Antarctica. The daily amount, type and stable water isotope ratios of precipitation have been monitored for about 3 years (2008-2010) at dome C. Using atmospheric models, the synoptic conditions generating precipitation at dome C are clustered and analyzed. Back trajectories are also computed to identify the dominant sources of moisture leading to precipitation at the considered location. A simple fractionation model is run but the simulated isotope ratio values are biased and poorly compare to local measurements. These results question the usual assumption related

to the interpretation of isotope ratio values in ice cores, and motivate further research to better understand the complex mechanisms governing precipitation and isotope fractionation over Antarctica.

**Recommendation**

This manuscript presents original and useful data, and raises important questions about the interpretation of isotope ratios for climate perspectives. The data and methods are well described and seem solid. I have no major issue, and I hence recommend to send the manuscript back to the authors for minor revisions. I have some comments and suggestions, listed below.

**General comments**

1. The fact that the precipitation measurements are not reliable in case of (relatively) strong wind is mentioned (e.g. p.14, l.5-6) but its influence on the presented analyses and results is not discussed. It would be instructive to provide the frequency of such "windy precipitation events" so the reader can figure if it is only marginal or on the contrary quite usual.

2. the organization of the paper is a bit strange (at least to me): the introduction is quite short, and previous work is discussed in Section 3 (I would expect this in the introduction). The limitation of the present organization is that the motivation for the present work, given in the introduction, is a bit weak because not put in the more general context presented later on. Up to the authors...

3. The authors must make an effort to explain the new contribution of the present

work with respect to previous studies by some of the authors (ex: Dittmann et al., ACP, 2016; Schlosser et al, ACP, 2016).

**Specific comments**

1. P.2, l.8: models do not provide data, but simulations (potentially constrained by observations).

2. P.10, l.13-24: in mixed-phase clouds where ice and liquid water particle co-exist, the Bergeron-Findheisen process is one possible mechanisms, but riming could also take place with very different cinematic (and involving collision). Could riming have different influence on fractionation?

3. P.10, l.30: I suggest to use "positively skewed distribution" rather than "L-distribution".

4. P.11, l.26: how this classification was conducted? this is an important method-ological aspect that must be clarified for the repeatability of the work.

5. P.12, l.6: "considerable amount": please provide numerical values. For readers not very familiar with Antarctica, the numbers may seem quite low...

6. P.15, l.1: Figure 10 is referred to in the text before Figure 9.

7. P.15, l.6-7: how (and why) were the events selected for the computation of the back trajectories?

8. Figure 1: it seems that the y-axis correspond to the number of occurrence rather than the frequency.

9. Figure 10: the legend should be moved in the plot to avoid masking points.

---

## Author Comment (AC2) · 17 May 2017

Final response to Referee 1

We appreciate Referee #1's effort to provide a review within short time. However, we got the impression that several things have been overlooked or understood incompletely.

This manuscript presents published measurements of d18O and d-excess of precipitation at the Dome C site as well as an application of a modeling approach combining the mesoscale atmospheric model and a simple isotopic model. The authors conclude that the model underestimates the depletion of d18O in precipitation in Antarctica. This

study does not provide any new data, despite the fact that the "Precipitation and stable isotopes data" are not presented in the part "Previous work". Everything has already been published in the paper by Stenni et al. (The Cryosphere, 2016).

It is true that a large part (not all!) of the data used in this study are already published, however, they have never been combined in the way we did it in the presented manuscript and were also supplemented by additional data. In order to make a publication self-contained, it is almost always necessary to explain some things that have been published before. We would like to stress that our study does yield new results that have not been published elsewhere. Stenni et al. (2016) stress the relationship of stable isotope ratios with meteorological station data. They discuss in detail the ïĄd'-T slope for various time periods and isotope variables, compare this to other locations and also look at the relationships amongst the isotope variables. The general atmospheric flow conditions are discussed only briefly, whereas in the new study we present a detailed analysis of the various synoptic situations that lead to precipitation at Dome C. Stenni et al. (2016) also state that hoar frost has a distinct fingerprint among the various precipitation types, implying that moisture sources and or the hydrological cycle might be different for hoar frost. Our more detailed study showed that this "fingerprint" is due to the fact that hoar frost occurs predominantly during the cold period. Relatively large amounts of hoar frost are measured after synoptic snowfall events, when humidity is still increased after moisture transport from lower latitudes, which means that hoar frost basically has the same moisture sources as the other precipitation types. The results on Dome C meteorological conditions and synoptic patterns during precipitation are already largely shown and discussed in the paper by Schlosser et al. (ACP, 2016).

In Schlosser et al. (2016), the stable isotopes served mainly as motivation for the study. They only discussed the meteorological conditions in two extreme years, without any isotope modelling or specific discussion of the stable isotope data and without any general analysis of the synoptic conditions during precipitation events. For instance, the conditions shown in Fig. 4d and 4e did not occur in the analysis of 2009 and

2010. Especially the situation in Fig 4e is highly interesting due to its relation to the Amundsen-Bellingshausen Sea Low. Nothing comparable occurs at Dome Fuji, so it was not discussed in the study by Dittmann et al. (2016).

The results of the isotopes modelling (part 5.4) have already been discussed largely by Dittmann et al. (ACP, 2016) for the Dome F site with the same conclusion. I thus do not see the added value of this study which is basically only a second application of the Dittmann et al. (2016) study on another site with similar characteristics. The main conclusion of this paper and of Dittmann et al. (2016), i.e. that the MCIM does simulate too high d18O in Antarctic precipitation is not new.

Dittmann et al. (2016) used a very short time series (less than 1 yr) from a different Antarctic location to study synoptic conditions and model stable isotopes. Even if we had done only the same for Dome C, it would be a valuable result to confirm Dittmann's findings with a longer time series from another location. Additionally, we used radiosonde data (not available for Dome Fuji for Dittmann's study) to determine the temperature at the lifting condensation level (LCL). This temperature, additionally to the temperature at the upper limit of the inversion layer, was used as input for the isotope model. It is a surprising result that this did not improve the model simulations. It is a very critical point for the relationship between temperature and stable isotope ratio, WHICH temperature is considered here. For many years, the temperature at the top of the inversion layer has been used, which is a strong simplification and more research is needed here.

This has already been noted for example in Uemura et al. (CP, 2016).

There is no publication by Uemura et al., CP 2016 (as suggested by the referee) to be found on the CP homepage. We are not aware of any study by Uemera et al. that investigates/models data from single precipitation events.) We did include a study by Uemura et al. from JGR2008

I also feel that the introduction part is misleading with very few references to previous

studies while much has been done in the recent years on the study of precipitation patterns and water isotopic composition in sites of the Antarctic plateau. The 2 recent papers mentioned in the introduction refer to Greenland studies. Similarly, the conclusion is very poor and only rephrase conclusions from previous studies (Schlosser et al., 2016; Dittmann et al., 2016; Stenni et al., 2016) without anything more.

We included new references concerning Antarctic synoptics/precipitation and about stable isotope work in Antarctica and in the lab. We also re-structured the manuscript by combining the sections "Introduction" and "Previous work" and we also re-wrote the conclusions. (see also comments to Ref.#2)

Response to Referee 2

We would like to thank the reviewer for the thorough and constructive review. In the following we address all single points: General comments

1. The fact that the precipitation measurements are not reliable in case of (relatively) strong wind is mentioned (e.g. p.14, l.5-6) but its influence on the presented analyses and results is not discussed. It would be instructive to provide the frequency of such "windy precipitation events" so the reader can figure if it is only marginal or on the contrary quite usual.

We added information about the frequency of such events in the text and also about the possible consequences for our results. Comparison of the total precipitation amount derived from the sampling to data from an accumulation stake array yields that the amount of sampled precipitation is lower than the measured accumulation. However, accumulation is also influenced by the wind (during and after precipitation), so an exact comparison is not possible.

2. the organization of the paper is a bit strange (at least to me): the introduction is quite short, and previous work is discussed in Section 3 (I would expect this in the introduction). The limitation of the present organization is that the motivation for the

present work, given in the introduction, is a bit weak because not put in the more general context presented later on. Up to the authors...

We agree that the introduction is very general and we need the information given in Section 3 (Previous work) to explain our motivation in more detail. Thus we re-structured the manuscript by combining the sections "Introduction" and "Previous work"

We also included new references concerning Antarctic synoptics/precipitation and about stable isotope work in Antarctica and in the lab following the advice of Referee #1.

3. The authors must make an effort to explain the new contribution of the present work with respect to previous studies by some of the authors (ex: Dittmann et al., ACP, 2016; Schlosser et al, ACP, 2016).

We re-wrote the conclusions stressing the new achievements and also added an additional result in the abstract.

Specific comments

1. P.2, l.8: models do not provide data, but simulations (potentially constrained by observations).

This is a semantic question. "Data" is a very general term, not even restricted to science, and does not, by definition, mean observational data (otherwise the widely used term "observational data" would be a tautology. Most modellers (including ECMWF and other big modellers) call their products model data. We do not see any problem here.

2. P.10, l.13-24: in mixed-phase clouds where ice and liquid water particle co-exist, the Bergeron-Findheisen process is one possible mechanisms, but riming could also take place with very different cinematic (and involving collision). Could riming have different influence on fractionation?

Riming would involve freezing of supercooled droplets. There is a kinetic effect during freezing of liquid water. However, since the freezing occurs rapidly, the fractionation is so weak that the model assumes fractionation for this phase transition is negligible. We changed the text accordingly.

3. P.10, l.30: I suggest to use "positively skewed distribution" rather than "Ldistribution".

We changed this in the text.

4. P.11, l.26: how this classification was conducted? this is an important methodological aspect that must be clarified for the repeatability of the work.

The classification was done manually. This has the advantage that the investigator is in full control of the process and the classification system can be tailored precisely to the researcher's needs. We added this information in the text.

5. P.12, l.6: "considerable amount": please provide numerical values. For readers not very familiar with Antarctica, the numbers may seem quite low...

We fully agree. We added information about precipitation amounts in the text.

6. P.15, l.1: Figure 10 is referred to in the text before Figure 9.

We exchanged the Figure numbers.

7. P.15, l.6-7: how (and why) were the events selected for the computation of the back trajectories?

Back trajectories were calculated for all cases where the synoptic situation seemed to be suitable for it, meaning a rather clear atmospheric flow. When this was not the case, trajectories tended to have kinks and loops and were not plausible or convincing. Again, this choice was made manually. We added this information in the text.

8. Figure 1: it seems that the y-axis correspond to the number of occurrence rather than the frequency.

We changed this in the figure to "number of observations".

9. Figure 10: the legend should be moved in the plot to avoid masking points. Done We also noticed that we had forgotten the reference Ciais and Jouzel, 1994 (and, incorrectly had used (Ciais et al., 1994) in the text. We corrected this and added the reference in t

We add the revised version with the corrected figures as supplement.

Please also note the supplement to this comment:
http://www.the-cryosphere-discuss.net/tc-2017-21/tc-2017-21-AC2-supplement.pdf

**Supplement:**

[revised manuscript text omitted]

b) weak anticyclone with northwesterly flow

c) anticyclone with northeasterly flow 15 Mar 2007 00Z

d) splitting of flow

e) southerly flow from West Antarctica

[Figure]

[Figure]

f) post event

[Figure]

[Figure]

**Figure 4: Synoptic patterns classification**

500 hPa geopotential height (contour interval 10gpm) and 24h precipitation totals from AMPS archive for the different synoptic situations during precipitation:

5 a) blocking anticyclone with northwesterly flow (23 May 2007)

b) weak anticyclone with northwesterly flow (13 Feb 2007)

c) anticyclone with northeasterly flow (16 Mar 2007)

d) splitting of flow (14 Aug 2008)

e) southerly flow from West Antarctica (3 May 2007)

10 f) post event (28 May 2007)

500 hPa geopotential height fields stem from AMPS 12h forecast of the preceding day, corresponding to 00 UTC of the described day, precipitation is AMPS 12h-36h forecast of the preceding day, corresponding to 00-24 UTC of the day described.

[Figure]

**Figure 5: Sea level pressure from AMPS (domain 1) for 3 May 2007 00 UTC**

[Figure]

**Figure 6: Wind speed W_s and direction for snowfall events identified in AMPS data**

[Figure]

**Figure 7: Observed wind speed at AWS vs. observed and modelled 24h – precipitation at Dome C**

[Figure]

**Figure 8:**

a) $\delta^{18}$O of precipitation samples vs. 2m air temperature from AWS for the different types of precipitation, snow, diamond dust, and hoar frost. Cases, for which trajectories were calculated, are marked with circles.

b) $\delta^{18}$O of precipitation samples vs. deuterium excess

**a)**

[Figure]

**b)**

[Figure]

**Figure 9: Observed and modelled a) $\delta^{18}$O and b) deuterium excess for days with moisture source estimates with Dome C temperature taken at 500hPa level, 600hPa level, and at the upper limit of the temperature inversion layer (derived from radiosondes as described in the text). The green squares mark days, for which trajectory 10 calculations were carried out.**

[Figure]

[Figure]

**Figure 10: Estimated moisture source areas for arrival levels a) 600 hPa and b) 500 hPa. Stronger colour corresponds to higher frequency of occurrence of the respective moisture source.**

---

## Author Response (AR2)

**Response to Referee #1 (revised version)**

We would like to thank Referee #1 for the review of the revised version. In the following, we answer to the comments point by point.

A-
- All the water isotopic data have been published in the paper by Stenni et al., the cryosphere, 2016. This should be presented as such. As a consequence, there is no reason why the acquisition of the data should be detailed except if additional measurements are presented here. If this is the case, it should be clearly stated.

One thing typically sought by readers and reviewers is that the paper is self-contained. Our motivations here have been to make this paper, and its documentation of the methodology and key input data sets, sufficiently self-contained and to avoid forcing readers to read another paper while studying ours, if not necessary. We agree with the reviewers' concern of not unduly repeating information, so we have reduced the original description by deleting the paragraph about sample transport and analysis. So, in the re-revised version we think we've avoided too much repetition, while still serving our goal of providing the minimum necessary and convenient information about the data that are used in this study.

- The model approach has already been applied at Dome F by Dittmann et al. (2016). So that this paper shares many similarities with this recently published paper. The site is still not the same. Since the two sites are on the East Antarctic plateau but under different influences, a nice added value to the paper would have been to develop a comparison between the two sites both on meteorological and isotopic aspects.

As we state in the introduction now, it is highly valuable to have exactly the same methods used for calculation of transport pathways and isotopic fractionation as well as for synoptic analysis, as often past studies have site-specific approaches, making comparisons very challenging. We also added a paragraph with a comparison of Dome Fuji and Dome C.

- The weather dynamic at Dome C with warm intrusions of air has already been discussed in the recent paper by Schlosser et al. (2016) concentrating on the year 2009 and 2010 without isotopic data. The only new feature presented here is the evidence of a moisture transport from West Antarctica across the continent and the relation to the Amundsen-Bellingshausen Sea Low. This part is only discussed within 15 lines (section 4e). If this is the major new result, it should be discussed much more.

Schlosser et al. (2016) focused on explaining the differences of the meteorological conditions of the two extreme years 2009 and 2010. However, no systematic analysis of different synoptic situations was carried out and no consequences for stable isotope ratios of precipitation were discussed. Here, however, we do, and this we think has resulted in a broader understanding, one of our goals.

B- The discussion on hoar isotopic composition (stated as a prominent new result in the conclusion and in contradiction with Stenni et al., 2016) is not clear. Only a few sentences are given in p. 15:

 « *Figure 8a shows observed d18O vs. 2m air temperature for the different types of precipitation: snow, diamond dust, and hoar frost. High precipitation events, for which trajectories were calculated, are marked with circles. The regression lines differ only slightly for the various precipitation types. For all samples, a d18O/T slope of 0.49‰/°C is found (r=0.79, n=498). The slope for the studied high*

*precipitation events only is 0.39‰/°C, lower than for all days (r=0.78, n=21). Also, the relationship between deuterium excess and d18O (Fig 8b) shows no significant differences between the precipitation types."*

This statement is not different from the finding by Stenni et al. (2016) who also display the same d18O vs Temperature slope for the different precipitation types (Table 4, hence similar to Figure 8 of this study). In Stenni et al., 2016 as in this manuscript, it is already mentioned that hoar mainly occurs during winter and that this may be the cause of the low d18O and high d-excess.

The discussion of Stenni et al. was basically qualitative. Here, however, we have quantified the analysis, which goes beyond that previous work. The result, we think, provides a more precise view and understanding of the issue, and so is valuable to retain.

 The other possibility proposed in Stenni is the fact that hoar is linked to low level water vapor isotopic composition and that one needs to explore this aspect. The study presented here does not explore this aspect because it better looks at high level (500 hPa, 600 hPa) trajectory.
It does not bring any information on low level water vapor isotopic composition so that it cannot conclude anything on the post-deposition influence on hoar isotopic composition. It is not possible to really conclude here on the question raised by Stenni et al., 2016 without knowing what happens in the low level atmospheric layer.

At first glance the levels may sound high, but we note that the 600hPa level can not be called a high level (above ground level (AGL)) at Dome C. Dome C is situated at an elevation of 3233m a.s.l., and the monthly mean surface pressure at Dome C varies between 630 and 650hPa, with daily values being considerably lower (lowest observed surface pressure: 603.6hPa). 600hPa, therefore, is the lowest standard pressure level that is always above the surface. Thus, the flow at 600hPa is representative for the flow at lower levels above ground level (order of magnitude: 0-500m above the surface) and is an appropriate one for this analysis.
As we explain in the text, the moisture, which is also involved in the formation of hoarfrost, basically (apart from the local cycle) has to come from the ocean on similar pathways as that for the falling precipitation.
The microscale processes involved in sublimation-resublimation are not the topic of our paper, and we think the low temperatures at which hoarfrost occurs explains the largest part of the low stable isotope values.

Finally, I find it disturbing that the authors claim that they have different conclusions than in Stenni et al., 2016. Indeed, the author list is quite similar between this study and the Stenni very recent paper.

We agree to that. But, as background, the Stenni et al. (2016) paper has a long history of development, and our more detailed investigation of the hoar frost cases did not take place until after Stenni et al. had been published. We feel that the newer results do differ from the previous ones (as it happens in science), however, and are worthwhile to be published on their own as extensions of prior, separate work.

C- The discussion on inversion vs condensation temperature used for calculating the d18O of precipitation in polar regions has already be explored by Ekaykin and others (see for example: http://eprints2008.lib.hokudai.ac.jp/dspace/bitstream/2115/45456/1/LTS68suppl_022.pdf, bottom of p. 301 and reference to phD thesis with temperature profiles above Vostok). These authors also provide some interesting conclusions that should be confronted to the results given here.

Agreed, and we have now included the mentioned reference in our paper, and we have added a discussion of the corresponding results. However, we note that Ekaykin et al. (2009) did not investigate single precipitation events, so the results are not exactly comparable with ours.

D- Finally, the strong effect of local temperature in d-excess in East Antarctica has already been discussed by others, i.e. Uemura et al., Climate of the Past, 2012. It is a quite well known effect of distillation that decreases the slope (dδD/dδ18O) towards low temperature.

We quote Uemura et al. (2012) now. We here discuss the measured and modelled d-excess and its relationship to moisture source conditions for precipitation events, which is different from what Uemura et al. did, though.

We also did some language editing; however, those numerous but small changes are not marked in the marked version in order to improve legibility.

For the editor:

Following the informal assessment of a third referee we also made some additional changes as follows:

*More discussion of the isotopic modeling is required. The author state that they did not find a correlation between the modelled and observed d18O values when using the condensation temperature as an input to the model, without suggesting an explanation. May be it's an issue related to the tuning of the model? Which value do they use for the supersaturation vs temperature coefficient, etc.? I would like to see more discussion of the different synoptic situations. For example, does precipitation formed in different synoptic situation differ in isotopic content? What about isotope-temperature slopes for different synoptic patterns?*

We have now added more information and discussion in the text. We have also discussed additional error possibilities.

*I found several mistakes, overlooked by the reviewers. Just one example: the moisture source for DC is Indian, not Pacific ocean.*

We appreciate this review; thanks. We have corrected this.

**List of changes:**

- We partly re-wrote the "Introduction".

- We shortened the data section.

- We added "Comparison of Dome C and Dome Fuji"

- We added information about the modeling.

- We extended the discussion about hoarfrost.

- We added information about the choice of arrival levels for trajectories.

- We explained why we did not calculate slopes for different synoptic situations.

- We discussed reasons for the poorer agreement of modeled and observed isotope ratios when the condensation temperature rather than the inversion temperature was used as model input

- We partly re-wrote the "Discussion and Conclusion".

- We added the following references: Motoyama (2007), (Schlosser et al. (2004), Uemura et al., (2012)

- We did a general language editing.

[revised manuscript text omitted]

---

## Author Response (AR3)

Dear Joel,

we hereby present the final version of our manuscript:

The influence of the synoptic regime on stable water isotopes in precipitation at Dome C, East Antarctica.

We made the last required small changes referring to the difference of Dome C and Concordia Station, the length of the Dome C core, and one typo.

We would like to thank you for all your work.

Yours sincerely

Elisabeth